# Investigating the contribution of grown new particles to cloud condensation nuclei with largely varying pre-existing particles – Part 1: Observational data analysis

Xing Wei[1#], Yanjie Shen[1,2#], Xiao-Ying Yu[3*], Yang Gao[1,4], Huiwang Gao[1,4], Ming Chu[1], Yujiao Zhu[5], Xiaohong Yao[1,4,6*]

[1]Key Laboratory of Marine Environment and Ecology (MoE) and Frontiers Sci Ctr Deep Ocean Multispheres & Earth, Ocean University of China, Qingdao, China

[2]College of Biology and Oceanography, Weifang University, Weifang, China

[3]Materials Science and Technology Division, Oak Ridge National Laboratory, Oak Ridge, TN 37831-6136, USA

[4]Laboratory for Marine Ecology and Environmental Sciences, Qingdao National Laboratory for Marine Science and Technology, Qingdao, China

[5]Environment Research Institute, Shandong University, Qingdao 266237, China

[6]Sanya Oceanographic Institution (Ocean University of China), Yazhou Bay Science & Technology City, Sanya, China

#*Equally contributed to the study*; *Correspondence to*: Xiao-Ying Yu (yuxiaoying@ornl.gov) and Xiaohong Yao (xhyao@ouc.edu.cn)

**Abstract.** This study employed multiple techniques to investigate the contribution of grown new particles to the number concentration of cloud condensation nuclei (CCN) at various supersaturation (SS) levels at a rural mountain site in North China Plain from 29 June to 14 July 2019. On eight new particle formation (NPF) days, the total particle number concentrations ($N_{cn}$) were $8.4 \pm 6.1 \times 10^3$ cm$^{-3}$, which were substantially higher compared to $4.7 \pm 2.6 \times 10^3$ cm$^{-3}$ on non-NPF days. However, the $N_{ccn}$ at 0.2 % SS and 0.4 % SS on the NPF days were significantly lower than those observed on non-NPF days ($p < 0.05$). This was due to the lower cloud activation efficiency of pre-existing particles resulting from organic vapor condensation and smaller number concentrations of pre-existing particles on NPF days. A case-by-case examination showed that the grown new particles only yielded a detectable contribution to $N_{ccn}$ at 0.4 % SS and 1.0 % SS during the NPF event on 1 July 2019, accounting for $12 \pm 11$ % and $23 \pm 12$ % of $N_{ccn}$, respectively. The increased $N_{ccn}$ during two other NPF events and at 0.2 % SS on 1 July 2019 were detectable, but determined mainly by varying pre-existing particles rather than grown new particles. In addition, the hygroscopicity parameter values, concentrations of inorganic and organic particulate components, and surface chemical composition of different sized particles were analyzed in terms of chemical drivers to grow new particles. The results showed that the grown new particles via organic vapor condensation generally had no detectable contribution to $N_{ccn}$, but incidentally did. However, this conclusion was drawn from a small size of observational data, leaving more observations, particularly for long-term observations and the growth of pre-existing particles to the CCN required size, needed for further investigation.

**Keywords:** New particles; pre-existing particles; cloud condensation nuclei; hygroscopicity parameter; ammonium nitrate.

**1 Introduction**

       Global warming and climate change have become significant topics of discussion and debate in recent decades (IPCC, 2021). Atmospheric aerosols, activated as cloud condensation nuclei (CCN) at super saturation (SS) conditions, can form cloud droplets and provide substantial global cooling, subsequently alleviating warming (Twohy 2005; Dusek et al., 2006; Small et al., 2009; Kerminen et al., 2012, 2018; Sullivan et al., 2018; Zaveri et al., 2021; Yu et al., 2020). However, compared to other climate forcing factors, aerosol–cloud interactions remain the largest uncertainty (IPCC, 2021). The CCN may act as one of the key contributors to this uncertainty, considering their roles in increasing the number concentration of cloud droplets and decreasing the size of cloud droplets (Twomey et al., 1977; Albrecht, 1989; Andreae and Rosenfeld, 2008; Bulgin et al., 2008). To minimize the large uncertainty on aerosol–cloud interactions, significant efforts have been made to improve the understanding of CCN regarding their primary and secondary sources, and aerosol activation properties in different size ranges with varying SS from the polluted atmosphere to the remote clean atmosphere (Yu and Luo, 2009; Kerminen et al., 2012, 2018; Li et al., 2017a; Williamson et al., 2019; Zhu et al., 2019a; Iwamoto et al., 2021; Zaveri et al., 2021, Gong et al., 2022).

       New particle formation (NPF) refers to the phenomenon of gas–particle nucleation followed by the growth of newly formed particles in the atmosphere. During an NPF event, particle number concentrations (*PNCs*) can rapidly increase in different size ranges (Kulmala et al., 2004; Chu et al., 2019; Lee et al., 2019). The grown new particles have been reported to contribute to the budget of CCN based on field measurements and modeling studies (Kuang et al., 2009; Kerminen et al., 2012; Ehn et al., 2014; Kalivitis et al., 2015; Leng et al., 2014; Ma et al., 2016; Tröstl et al., 2016; Gordon et al., 2017; Li et al., 2017b; Williamson et al., 2019; Fang et al., 2021; Ren et al., 2021; Sebastian et al., 2021). However, field studies are often affected by perturbations associated with varying pre-existing particles in *PNCs*, size spectra, and cloud activation properties. As a result, observational evidence on the net contribution of the grown new particles to CCN concentrations is limited, particularly at low ambient SS, i.e., SS < 0.2–0.3 % (Kerminen et al., 2012; Tröstl et al., 2016; Rose et al., 2017; Ren et al., 2021; Sebastian et al., 2021).

       Organic vapors have been found to play a dominant role in the growth of newly formed particles, ranging from a few nanometers to submicron scales (Pierce et al., 2012; Wu et al., 2013; Ehn et al., 2014; Kalivitis et al., 2015; Kawana et al., 2017; Ma et al., 2021; Wan et al., 2020; Chang et al., 2022). This organic-driven growth may reduce the Kelvin effect for water vapor condensation on small-sized particles (Dusek et al., 2006). Considering that the global average of the hygroscopicity parameter ($\kappa$) values of atmospheric aerosols is around 0.27 (Petters and Kreidenweis, 2007; Kerminen et al., 2012), the condensation of low-volatile organic vapors on grown new particles and pre-existing particles might lead to a decrease in aerosol hygroscopicity due to their low $\kappa$ values, which approach 0.1 (Petters and Kreidenweis, 2007; Dusek et al., 2010; Wu et al., 2013; Zhu et al., 2019b; Fang et al., 2021; Chang et al., 2022). However, some semi-volatile organic species, such as oxalic acid, have been found to have high cloud activation potentials, with a $\kappa$ value of 0.81, and their neutralized salts also have $\kappa$ values of 0.59–0.70 (Boreddy and Kawamura, 2018). Additionally, secondary organic aerosols (SOA) from dark ozonolysis of γ-terpinene under different reaction conditions have shown moderate cloud activation

potentials, with $\kappa$ values of 0.20–0.24 (Bouzidi et al., 2022). During an NPF event, when different organic vapors condense simultaneously on grown new particles and pre-existing particles, the $\kappa$ values of different sized particles can either decrease, increase, or remain invariant, leading to corresponding changes in CCN concentrations. Therefore, statistical analysis is essential in understanding the complex interactions between organic vapors and particle hygroscopicity, which is currently scarce in the literature.

Various studies have reported that sulfuric acid vapor generated from photochemical reactions in the atmosphere contributes to the growth of newly formed particles (Birmili et al., 2003; Boy et al., 2005; Yue et al., 2011, Bzdek et al., 2012, Vakkari et al., 2015; Ma et al., 2016; Fang et al., 2021). However, the contribution of sulfuric acid vapor to growing secondary particles from 40–50 nm to larger sizes is highly questionable, as observed subsequent growth often occurs at nighttime (Man et al., 2015; Ma et al., 2021). Furthermore, the relative contribution of ammonium nitrate ($NH_4NO_3$), one of the most important components in atmospheric particles, increases with a faster decrease in $SO_2$ emissions globally (Chan and Yao, 2008; Cao et al., 2017; Ge et al., 2017; Rodelas et al., 2019; Wu et al., 2019; Bressi et al., 2021; Zhu et al., 2021a). Current modeling studies suggest the importance of $NH_4NO_3$ aerosols in regional cooling (Drugé et al., 2019; Jones et al., 2021). Our previous studies have found that the formation of particulate $NH_4NO_3$ via $HNO_3$ and $NH_3$ gases plays a key role in the second-phase growth of newly formed particles from 40–50 nm to larger sizes at nighttime without photochemical reactions (Zhu et al., 2014; Man et al., 2015; Ma et al., 2021). Recently, Wang et al. (2020) proposed rapid particle formation and growth of newly formed particles by $NH_4NO_3$ formation. Unlike organic vapors, $NH_4NO_3$ formation on newly formed particles and pre-existing particles should always yield a net increase in CCN concentrations during NPF events. However, to the best of our knowledge, no field studies have reported this phenomenon.

The objective of this study is to investigate the relationship between newly formed particles and CCN using particle number size distributions ($PNSDs$), CCN concentrations, and the derived $\kappa$ values of atmospheric aerosols at various SS. Chemical composition of atmospheric particles was also analyzed by multiple techniques to support the analysis. The observations were conducted at a rural mountain site (1100 m above sea level) in North China Plain (NCP) from June to July in 2019, during the rainy season when morning mist frequently occurred among mountain peaks with ambient relative humidity (RH) approaching 90 %–100 % and decreasing ambient temperature (T) (Fig. S1a–b). Additionally, cloud optical thickness ($COT$) and cloud effective radius ($CER$) were recorded in the midafternoon using Aqua satellite data, showing variations from 1.46 to 36.7 and 8.64 μm to 24.4 μm, respectively (Fig. S1c–d). The observed CCN were directly related to the formation of cloud droplets at the elevated mountain site. Three scientific questions were addressed: (1) Does the organic-driven growth of newly formed particles contribute to the observed CCN at various SS? Is the effect of organic condensation reduced on pre-existing particles during NPF events? (2) What is the relative role of organic condensation and $NH_4NO_3$ formation in determining the contribution of grown new particles to the observed CCN at various SS? (3) What implications do our findings have on knowledge gaps for CCN sources in NCP? In a companion paper, a modeling study is applied to illustrate the link in three dimensions and quantify the contributions of different chemical species to the growth of newly formed particles (Chu et al., 2023).

## 2 Experimental

### 2.1 Sampling period and site

A research campaign was performed from 23 June to 14 July 2019 at a mountain site (115°26′ E, 39°58′ N) in western Beijing to investigate the CCN activation of atmospheric aerosols in the rural atmosphere over the North China Plain (NCP). The sampling site, located at an altitude of approximately 1100 m (Fig. 1a–f), belongs to a forest ecosystem research station of the Chinese Academy of Science. The mountain site is surrounded by secondary forests and is approximately 110 km southwest downwind from Beijing. $NO_2$ column densities during the study period showed that anthropogenic combustion emission sources were mainly distributed in the east and south of the site (Fig. 1a–f), while the north and west directions frequently displayed lower $NO_2$ column densities, indicating a cleaner rural environment.

### 2.2 Particle collection

Four different instruments were installed on the third floor of the station major building, at a height of 10 m above ground. These instruments include a Fast Mobility Particle Sizer (FMPS, TSI, 3091) downstream of a dryer (TSI, 3062), a Condensation Particle Counter (CPC, TSI, 3775), a continuous flow CCN counter (CCNC, DMT Model 100), and a Differential Mobility Analyzer (DMA, Grimm) coupled with a Nanometer Aerosol Sampler (NAS, Grimm). The FMPS was used to measure particle concentrations in the size range of 5.6 nm to 560 nm in 32 channels at a frequency of 1 Hz. For the purpose of this study, the total particle number concentration (total $N_{cn}$) and number concentrations of particles larger than 100 nm ($N_{cn>100}$) were defined as the sum of number concentrations from 5.6 nm to 560 nm and 100 nm to 560 nm, respectively. The FMPS had undergone maintenance at the TSI factory in the U.S. before the campaign, and the ratios of the measured total particle number concentrations against those measured by the CPC remained stable until the end of 2021. The CPC shared a splitter with the FMPS and generated data at a 2-second time resolution to correct the FMPS data (Zimmerman et al., 2015). The CCNC was used to measure the bulk CCN concentration ($N_{ccn}$) at five different supersaturation (SS) levels of 0.2 %, 0.4 %, 0.6 %, 0.8 %, and 1.0 %. Each SS setting lasted for 5 minutes, while the first and last 30 seconds of data were removed. An additional 5 minutes were used for switching SS from 1.0 % to 0.2 % to establish supersaturation equilibrium. The CCNC was calibrated for the campaign using the commercial service provided by the instrument vendor in Beijing. However, the SS were referred as the lab-calibrated values in this study since no on-site calibration was conducted during the campaign. The divergence in height between the observational site and the calibration location (~1000 m) may introduce uncertainties on SS and the consequently measured $N_{ccn}$ (Lance et al., 2006; Rose et al., 2008; Lathem and Nenes, 2011). Based on these previous studies, the on-site five SS at the mountain site might be approximately 10 % smaller than the corresponding lab-calibrated values, i.e., 0.18 %, 0.36 %, 0.54 %, 0.72 % and 0.9 %. These smaller SS values were referred as the approximated on-site values in this study. Moreover, the errors in the measured $N_{ccn}$ between each pair of lab-calibrated and approximated on-site SS were smaller than 10 % as presented in Supporting Information (Figs. 3–5 and Fig. S6). Nevertheless, these analytical errors are less likely to affect the comparison of $N_{ccn}$ at different times within a single day, as it can be reasonably assumed that the percentage of analytical

errors remains constant. Figure S2 showed the sampling system diagram. Further details about these instruments can be found in Li et al. (2015), Wang et al. (2019), and Gao et al. (2020) and the inter-comparison results of these particle-number-related measurements during the campaign and afterwards were presented in the Supporting Information and Figs. S3–S5.

The DMA coupled with the NAS was used to collect atmospheric nanoparticles at four diameters: 30 nm, 60 nm, 100 nm, and 200 nm. The DMA initially separated atmospheric particles, which were then collected on Transmission Electron Microscopy (TEM) grids (01814-F, TED PELLA, INC.) by the NAS using electrostatic force. To ensure that sufficient particles were collected for chemical analysis, each sample was collected for 2 hours, with a sampling rate of 1 L min$^{-1}$ and a sheath flow rate of 10 L min$^{-1}$. A total of 14 sample sets and three field blanks were collected. The chemical composition of the samples collected on 30 June and 1 July 2019 was successfully analyzed. Additionally, field blanks were analyzed to identify interference peaks from substrates.

During the campaign, a total of 11 total suspended particle (TSP) samples were collected from 06:00 on that day to 06:00 on the next day. These samples were collected on quartz filters using a high-volume sampler, which operated at a flow rate of approximately 1 m$^3$ min$^{-1}$. The collected samples were utilized for analyzing inorganic ions, organic carbon (OC), elemental carbon (EC), and organic tracers. The sampling duration of each sample was around 24 hours, which ensured sufficient particle loading for multiple chemical analyses. Additionally, meteorological data such as wind speed (WS), wind direction (WD), ambient temperature (T), and relative humidity (RH) were continuously measured at a height of 10 m above ground level at the station.

## 2.3 Chemical analysis of particles

A ToF-SIMS (time-of-flight secondary ion mass spectrometer) instrument (IONTOF V, IONTOF GmbH, Münster, Germany) was utilized to analyze the chemical composition of atmospheric nanoparticles collected using the DMA coupled with NAS at four different sizes. During analysis, the main chamber pressure was kept under a vacuum of 10$^{-8}$ mbar. The primary ion beam was Bi$^{3+}$ (25 keV) with 10 kHz pulse energy, a pulse width of 0.8 ns, and a current set at approximately 0.6 pA. The SIMS spectra were acquired over an area of 150 × 150 μm$^2$ for 60 scans, with at least 4 positive and 4 negative data points collected for each sample. The IONTOF Surface Lab 6.3 software was used for SIMS data analysis. The mass spectra were calibrated using $m/z^+$ 15 CH$_3^+$, 55 C$_4$H$_7^+$, and 91 C$_7$H$_7^+$ in the positive ion mode, and $m/z^-$ 26 CN$^-$, 41 C$_2$HO$^-$, and 77 CHO$_4^-$ in the negative ion mode, respectively. Principal component analysis (PCA) was conducted using MATLAB software to study the differences in chemical composition among the selected-size particles. Prior to conducting spectral PCA, the interference peaks from substances were deleted, and the mass-calibrated data were treated by mean-centering, normalization to the total ion intensity of all selected peaks, and square-root transformation (Ding et al., 2016, 2019; Zhang et al., 2019; Sui et al., 2017; Fu et al., 2018; Cheng et al., 2014). It should be noted that inorganic signals with weak signal-to-noise ratios were excluded from the analysis.

For the analysis of TSP samples, a 1/4 fraction of each filter was utilized to measure organic tracers using gas chromatography mass spectroscopy (GC-MS) with an Agilent 6890 GC/5975 MSD, as described in previous studies (Kleindienst et al., 2007; Feng et al., 2013). The OC and EC concentrations

were analyzed using a thermal/optical carbon analyzer (DRI 2001A, Atmoslytic Inc., USA) with the IMPROVE temperature program. First, the OC was volatilized under helium by heating the filter to 580 °C in four steps and then to 870 °C in three steps in a He: O$_2$ environment. The charring of OC was monitored using a He-Ne laser of 650 nm. The secondary OC (SOC) was estimated as (OC – 2 × EC) following the method of Yu et al. (2009). Inorganic and organic ions were analyzed using ion chromatography (Dionex 3000) from a 1/4 fraction of each TSP sample filter (Hu et al., 2015; Teng et al., 2017). More experimental details can be found in the Supporting Information.

**2.4 Definition of NPF events and calculation methods**

NPF events were identified using the criteria proposed by Dal Maso et al. (2005) and Kulmala et al. (2012), which require the observation of new nucleation mode particles in the spectra and their prevalence or growth over several hours. These criteria distinguish NPF particles from plume and pre-existing particles. NPF events were considered to have ended when the new particle signals disappeared, and the total particle number concentration returned to the background levels before the NPF events or when the newly formed particles were suddenly overwhelmed by plumes and could no longer be identified. NPF days (non-NPF days) were defined as the presence (absence) of NPF events. Two metrics, the apparent formation rate (*FR*) and the net maximum increase in the nucleation mode number concentration (*NMINP*), were used to quantify the intensity of NPF events (Sihto et al., 2006; Kulmala et al., 2012; Zhu et al., 2021a). *FR* and *NMINP* during the NPF events were calculated based on the nucleation mode particles in the size range of 5.6 nm to 30 nm. Particle number size distributions (*PNSDs*) in this study were fitted using multi-lognormal distribution functions in which the median mode mobility diameter and its corresponding half width (standard deviation) were used to characterize the distribution (Whitby, 1978; Yao et al., 2005; Chen et al., 2023). The related calculation methods are described in detail in the Supporting Information.

The $\kappa$ was calculated in Eq. (1) that proposed by Petters and Kreidenweis (2007):

$$\kappa = \frac{4A^3}{27D_d^3 \ln^2 S_c} \ , \ A = \frac{4\sigma_{s/a} M_w}{RT\rho_w} \tag{1}$$

where $D_d$ is the particle dry diameter, and it was assumed to be equal to the critical diameter for CCN activation ($D_{crit}$). $S_c$ is the supersaturation. $\sigma_{s/a}$ is set as 0.0072 J m$^{-2}$, representing the surface tension over the interface of solution and air, $M_w$ is the molecular of water, $R$ is the universal gas constant, $T$ is temperature, and $\rho_w$ is water density. In this study, the hourly $\kappa$ values at 0.2 %, 0.4 % and 1.0 % SS were calculated for analysis. $D_{crit}$ is defined as the particle diameter which is considered as the lower limit of the integral on particle number, and the upper limit was set as the largest particle diameter, yielding the total integrated particle number concentration equal to the CCN concentration (Hung et al., 2014; Cheung et al., 2020; Gao et al., 2020). The calculated hourly $\kappa$ values reflect the overall effect on particle size, chemical composition, and mixing state, which are thereby referred to as the bulk $\kappa$ values under different SS conditions.

Satellite products do not provide accurate retrievals of cloud droplet number concentration (*CDNC*) due to uncertainties introduced by lidar-derived biases. Building on the "adiabatic cloud model" assumption (Bennartz, 2007; Bennartz and Rausch, 2017), an effective computational method is

established to estimate liquid-phase *CDNCs* based on cloud optical thickness (*COT*) and cloud effective radius (*CER*) in Eq. (2) and Eq. (3):

$$CDNC = \frac{\tau^3}{k}[2W]^{-\frac{5}{2}}[\frac{3}{5}\pi Q]^{-3}[\frac{3}{4\pi\rho_w}]^{-2}c_w^{\frac{1}{2}} \tag{2}$$

$$W = \frac{5}{9}\rho_w\tau r_e \tag{3}$$

where $\tau$ is cloud optical thickness, $k$ equal to 0.8 presents the dispersion of the assumed cloud droplet size distribution, $Q$ is approximately 2 as the scattering efficiency of droplet, $c_w$ is the condensation rate and it is calculated following previous studies (Ahmad et al., 2013; Grosvenor et al., 2018; Li et al., 2018). $W$ is liquid water path (*LWP*), and $r_e$ is cloud effective radius. *COT* and *CER* were derived from daily mean level-3 daily 1° cloud retrieval product (MYD08_D3) retrieved from Aqua-MODIS and the dataset was from June 2019 to July 2019. The Aqua satellite scanned the cloud coverage over the observational site within the time range of 12:30 and 14:30 in each day.

We modeled 24-hour air mass back trajectories at 1000 m by using the Hybrid Single-Particle Lagrangian Integrated Trajectory (HYSPLIT) model from the NOAA Air Resources Laboratory. Simulations were performed every 2 hours starting from 2 hours before the NPF and ended until the NPF signals disappeared (Fig. 1a–f). $NO_2$ column densities were downloaded from https://so2.gsfc.nasa.gov/no2/no2_index.html.

## 3 Results and discussion

### 3.1 Overview of the measured $N_{ccn}$ related to NPF events

Figure 2a showed the hourly average $N_{ccn}$ at the lab-calibrated 0.2 %, 0.4 %, and 1.0 % SS and the 1-minute average total $N_{cn}$ from 29 June to 14 July 2019, respectively. Table S1 provides the daily average $N_{ccn}$ (average ± standard deviation) at SS values ranging from 0.2 % to 1.0 %, the daily average $N_{cn>100}$, and the total $N_{cn}$. The $N_{ccn}$ varied by approximately 1 order of magnitude, ranging from $0.10 \times 10^3$ cm$^{-3}$ to $4.7 \times 10^3$ cm$^{-3}$ ($1.4 \pm 0.7 \times 10^3$ cm$^{-3}$) at 0.2 % SS and from $0.10 \times 10^3$ cm$^{-3}$ to $4.8 \times 10^3$ cm$^{-3}$ ($1.7 \pm 0.9 \times 10^3$ cm$^{-3}$) at 0.4 % SS. The temporal trend in $N_{ccn}$ at 1.0 % SS was consistent with those at 0.2 % and 0.4 % SS, while the values of $N_{ccn}$ at 1.0 % SS were $2.2 \pm 1.1 \times 10^3$ cm$^{-3}$, which was approximately 60 % higher than that at 0.2 % SS. The $N_{ccn}$ values observed on 29–30 June 2019 were $0.69 \pm 0.27 \times 10^3$ cm$^{-3}$, $0.83 \pm 0.36 \times 10^3$ cm$^{-3}$, and $1.2 \pm 0.6 \times 10^3$ cm$^{-3}$ at 0.2 %, 0.4 %, and 1.0 % SS, respectively. These values decreased by approximately 50 % relative to the campaign averages observed during the 16 days. However, the total $N_{cn}$ on those 2 days ranked at a moderately high level (see Table S1), leading to lower activation ratios (*ARs*) of the observed aerosols, i.e., $0.12 \pm 0.07$ at 0.2 % SS, $0.15 \pm 0.09$ at 0.4 % SS, and $0.20 \pm 0.13$ at 1.0 % SS (Fig. 2b). Although no size-dependent chemical components were measured, the calculated ratios of SOC/ ($SO_4^{2-}$ + $NO_3^-$) in TSP on those 2 days were substantially larger than those from other days (see Fig. 2c), implying that the larger fraction of organics prevented aerosols from being activated as CCN (Petters and Kreidensohler, 2007; Kawana et al., 2017; Zhu et al., 2019b; Crumeyrolle et al., 2021; Chang et al., 2022). Surprisingly, the satellite-derived *CDNC* of 341–386 cm$^{-3}$ over the mountain area were higher on those 2 days (see Fig. 2d). These values were higher compared to the

*CDNC* levels observed in June–July of 2019, which varied around $141 \pm 122$ cm$^{-3}$ (fig superimposed in Fig. 2d). On the other 14 days, the *CDNC* were determined to be $106 \pm 55$ cm$^{-3}$, approximately 1 order of magnitude smaller than the observed corresponding to $N_{ccn}$ at 0.2 % SS. The difference between the observed $N_{ccn}$ and the *CDNC* implied that the SS required for cloud droplet formation in the NCP atmospheres might be substantially smaller than 0.2 %, even though $N_{ccn}$ may decrease to some extent at higher altitudes from 1000 m (Li et al., 2019; Yang et al., 2020; Che et al., 2021). Thus, we will further explore this difference in Sect. 3.5. Note that the estimation *CDNC* might suffer from the uncertainty to some extent because of complex microphysics of cloud (Pruppacher and Klett, 2012). However, this uncertainty is unlikely to have significantly affected the comparison results of *CNDC* with $N_{ccn}$ since the latter difference generally exceeded 1 order of magnitude.

Compared to observations at 0.2 % SS reported in the literature, the campaign average $N_{ccn}$ during the 16 days in this study was almost double the value of $0.8 \pm 0.7 \times 10^3$ cm$^{-3}$ measured during 3 weeks of summer in 2014 on Mt. Huang (Anhui province, China) (Miao et al., 2015) and $0.7 \pm 0.4 \times 10^3$ cm$^{-3}$ measured during 4 weeks of summer in 2019 on Mt. Tian (Xinjiang Uygur Autonomous Region, China) (Liu et al., 2020). In addition, even lower seasonal averages of $N_{ccn}$ at 0.2 % SS were recently observed on Liupan Mountain (Ningxia Hui Autonomous Region, China) from August 2020 to November 2021, with the lowest seasonal average being $0.4 \times 10^3$ cm$^{-3}$ in summer and the highest being $0.7 \times 10^3$ cm$^{-3}$ in winter (Lin et al., 2022). However, the average $N_{ccn}$ was only half of that observed at a suburban site of Qingdao (Shandong province, China) in the spring of 2013 (Li et al., 2015). Schmale et al. (2018) also analyzed $N_{ccn}$ at 0.2 % SS observed at 12 regionally representative sites, including two mountain sites, i.e., Jungfraujoch (JFJ) mountain station in Switzerland from January 2012 to December 2014, and Puy de Dôme (PUY) mountain station in France from November 2014 to September 2015. The recorded maximum values at the JFJ and PUY did not exceed $1 \times 10^3$ cm$^{-3}$. They also reported a notable seasonal variation in $N_{ccn}$ at these mountain sites, with the highest median concentrations occurring in summer associated with the uplifted boundary layer air masses. However, the summer median values were far smaller than $1.0 \times 10^3$ cm$^{-3}$, e.g., $0.2 \times 10^3$ cm$^{-3}$ at JFJ. At 0.4 % SS, the average $N_{ccn}$ during the 16 days was also double the values observed at SMEAR II station (a rural forest site) in Hyytiälä in July 2008 and June 2009 (Sihto et al., 2011), and substantially larger than the values of 166–700 cm$^{-3}$ at 0.11–0.80 % SS over a mid-latitude forest in Japan (Deng et al., 2018). Assuming that the rural seasonal or campaign averages of $0.3$–$0.8 \times 10^3$ cm$^{-3}$ $N_{ccn}$ at 0.2 % SS observed at these remote mountain sites in China represent upper limits for natural contributions, it is likely that at least 50 % of the observed $N_{ccn}$ in this study were derived from primary and secondary anthropogenic aerosols. In Beijing, NPF events have also been observed in polluted atmospheres with air masses originating from the south and southwest (Wu et al., 2007). In those cases, anthropogenic aerosols expectedly yield an even larger contribution to $N_{ccn}$. Moreover, the measured concentrations of EC in TSP were only 0.23–0.53 μg m$^{-3}$ in this study (see Table 2), consistent with values of 0.10–0.43 μg m$^{-3}$ in PM$_{2.5}$ in remote atmospheres across North America (Ahangar et al., 2021). Given the low CCN activation of primary aerosols (Gao et al., 2020), it is expected that secondary anthropogenic aerosols contributed significantly to $N_{ccn}$ (Ma et al., 2021) and were therefore investigated below.

Nucleation events have been identified as significant sources of atmospheric particles in terms of number concentration (Kulmala et al., 2004; Chu et al., 2019; Lee et al., 2019). In this study, such events were observed frequently on 8 out of the total 16 days (Figs. 3–5 and Fig. S7), with air masses mainly originating from the northwest, north, and northeast during these events (Fig. 1a–f). Shorter air mass back trajectories were obtained from 12 July to 14 July, and shorter durations of nucleation events were observed. Wu et al. (2007) reported statistical analysis of NPF events using a year measurement in Beijing, i.e., NPF events occurred under low RH and sunny conditions while non-NPF events were usually associated with strong condensational sink or absence of sunny conditions. The air masses from the north and northwest usually carry dry and clean air, favoring the occurrence of NPF events. As shown in Table S1, significantly lower $N_{ccn}$ values were observed on nucleation days compared to non-nucleation days (with $p < 0.05$), namely, $1.2 \pm 0.7 \times 10^3$ cm$^{-3}$ on nucleation days versus $1.6 \pm 0.8 \times 10^3$ cm$^{-3}$ on non-nucleation days at 0.2 % SS. The same trend was observed at 0.4 % SS, with the values of $1.5 \pm 0.9 \times 10^3$ cm$^{-3}$ on nucleation days and $1.8 \pm 0.9 \times 10^3$ cm$^{-3}$ on non-nucleation days, respectively. However, the $N_{ccn}$ values at 1.0 % SS on nucleation days ($2.1 \pm 1.2 \times 10^3$ cm$^{-3}$) did not differ significantly from those observed on non-nucleation days ($2.3 \pm 1.1 \times 10^3$ cm$^{-3}$) ($p > 0.05$). Hirshorn et al. (2022) analyzed 15 years of observational data from mountaintop location in North America and found that $N_{ccn}$ did not show a significant increase during NPF events in either summer or autumn. Kawana et al. (2017) also reported lower mean $N_{ccn}$ values on nucleation days compared to non-nucleation days at a forest site in Wakayama (Japan) during the summer, which were likely due to biogenic secondary organic condensation on atmospheric particles. A comprehensive survey on this issue reported in the literature will be presented in Sect. 3.4. Theoretically, newly formed particles may continue to grow even after the new particle signal disappears from observations. However, the occurrence of pre-existing particle growth seemed infrequent and was observed only on 4 July (1 of 16 days). On that day, larger and smaller pre-existing particle growth was observed from 88 nm to 116 nm and from 24 nm to 32 nm, respectively (see Fig. S8). Moreover, the growth of newly formed particles extended to the next day, occurring only during one nucleation event, where newly formed particles grew from 1 July to the early morning of 2 July. Therefore, the observations on 2 July were reclassified as nucleation days and removed from non-nucleation days. The re-calculated $N_{ccn}$ values on nucleation days did not differ significantly from those on non-nucleation days at SS = 0.2 % and 0.4 % with $p > 0.05$, namely, $1.3 \pm 0.5 \times 10^3$ cm$^{-3}$ (nucleation days) versus $1.4 \pm 0.5 \times 10^3$ cm$^{-3}$ (non-nucleation days) at 0.2 % SS, and $1.6 \pm 0.7 \times 10^3$ cm$^{-3}$ (nucleation days) versus $1.6 \pm 0.6 \times 10^3$ cm$^{-3}$ (non-nucleation days) at 0.4 % SS. The change suggested that the significance of the $N_{ccn}$ discrepancy between nucleation and non-nucleation days at 0.2 % and 0.4 % SS was sensitive to the applied data size. However, further investigation through long-term observations is required to determine whether nucleation events can contribute to a statistically significant increase in CCN concentration.

Excluding 2 July, the $N_{cn>100}$ on NPF days were significantly lower at ($1.6 \pm 0.8 \times 10^3$ cm$^{-3}$) compared to non-NPF days ($1.8 \pm 1.0 \times 10^3$ cm$^{-3}$) with $p < 0.05$ (Table S1), which partially explained the lower $N_{ccn}$ at 0.2 % SS. However, the total $N_{cn}$ substantially increased on NPF days to $8.4 \pm 6.1 \times 10^3$ cm$^{-3}$ compared to $4.7 \pm 2.6 \times 10^3$ cm$^{-3}$ on non-NPF days. During the first 2–3 hours of the NPF events, the total $N_{cn}$ increased significantly from $4.2 \pm 1.1 \times 10^3$ to $21 \pm 5.4 \times 10^3$ cm$^{-3}$ with the $NMINP$ of $17 \pm 6.0$

$\times 10^3$ cm$^{-3}$ (Fig. 2a). Nonetheless, the increase in the total $N_{cn}$ did not result in a statistically significant increase in the observed $N_{ccn}$ at the lab-calibrated SS from 0.2 % to 1.0 % on NPF days compared to non-NPF days, regardless of including or excluding 2 July from NPF days. It is important to note that the observational data alone cannot provide evidence of any additional evolution of grown new particles and

their additional contribution to $N_{ccn}$ after the new particle signal disappears, particularly considering the infrequent occurrence of the pre-existing particle growth.

The statistical comparisons between NPF and non-NPF days in this study showed contradictory results to those previously reported in the literature. In other words, the findings of this study contradicted the idea that NPF events are important secondary sources of CCN, which has been reported in previous

studies (Sotiropoulou et al., 2006; Kuwata et al., 2008; Wiedensohler et al., 2009; Sihto et al., 2011; Ma et al., 2016; Rose et al., 2017; Kerminen et al., 2018; Wan et al., 2020; Fang et al., 2021; Ren et al., 2021; Chang et al., 2022). Therefore, this paper presents a case-by-case examination from Sect. 3.2 to Sect. 3.5 where the contributions of NPF events to the $N_{ccn}$ are elaborated at various SS with considering the grown new particles in different sizes. Section 3.2 presents the analysis results of five NPF events on 29 June,

3 and 12–14 July. The NPF events did not cause an increase in $N_{ccn}$ at 0.2 %, 0.4 %, and 1.0 % SS (see Fig. 3 and Fig. S7). Section 3.3 includes two NPF events on 30 June and 6 July when an increase in $N_{ccn}$ was indeed observed (see Fig. 4). However, this increase was associated mainly with changing number concentrations and/or $\kappa$ values of pre-existing particles. On 1 July, an increase in $N_{ccn}$ was observed during the NPF event (see Fig. 5), while the grown new particles unlikely contributed to $N_{ccn}$ at 0.2 %

SS. The results are analyzed in the first part of Sect. 3.4. When the $N_{ccn}$ at 0.4 % and 1.0 % SS were analyzed in the NPF event on 1 July, the contributions of the grown new particles to $N_{ccn}$ could be reasonably qualified at nighttime with a large increase in $\kappa$ values of atmospheric particles. These results will also be included in the second part of Sect. 3.4.

**3.2 Case study with no detectable contribution to $N_{ccn}$ during NPF events**

Figure 3a–f shows that there was no detectable contribution of grown new particles to $N_{ccn}$ in the NPF events on 29 June and 3 July. Specifically, higher values of $N_{ccn}$ at the lab-calibrated 0.2 % and 0.4 % SS were observed before 06:00 (24 hours used here and afterwards) on 29 June, which were associated with the intrusion of aerosol plumes and higher $\kappa$ values of 0.26 ± 0.06 at 0.2 % SS and 0.16 ± 0.03 at 0.4 % SS (see Fig. 3a–c). The $\kappa$ values at 0.2 % SS were close to the global average values and those

reported in suburban or rural polluted atmospheres of China (Petters and Kreidensohler, 2007; Rose et al., 2010, 2011; Ma et al., 2016; Fang et al., 2021). Assuming that the activated aerosols at 0.2 % SS were internally mixed and mainly composed of inorganic ammonium salts and organics (Petters and Kreidensohler, 2007; Rose et al., 2010, 2011), both of them likely yielded an appreciable contribution to the total mass concentration of the associated aerosols. However, the exact percentages relied on the $\kappa$

values of various organics. The $\kappa$ values of 0.16 ± 0.03 at 0.4 % SS were also reported in remote forest or less polluted areas (Gunthe et al., 2009; Dusek et al., 2010; Cerully et al., 2011; Sihto et al., 2011; Levin et al., 2014; Kawana et al., 2017; Fang et al., 2021; Park et al., 2021). The calculated $\kappa$ values almost halved with SS increasing from 0.2 % to 0.4 % before 06:00, suggesting that the observed aerosols in smaller sizes had lower cloud activation potentials. Similar results were frequently reported in the

literature, i.e., the fraction of organics in atmospheric nanometer particles increased with the decrease of particle sizes (Rose et al., 2010, 2011; Crippa et al., 2014; Cai et al., 2017). After 06:00, the calculated $\kappa$ values at 0.2 % and 0.4 % SS largely decreased to be less than 0.1. The low $\kappa$ values suggested that atmospheric aerosols measured after 06:00 mainly consisted of low CCN-activated organics (Petters and Kreidensohler, 2007; Kerminen et al., 2018; Chang et al., 2022). When the approximated on-site SS were used to derive the $\kappa$ values before 06:00, they were $0.32 \pm 0.08$ at 0.18 % SS and $0.19 \pm 0.07$ at 0.36 % SS, respectively. After 06:00, they also decreased to be blow 0.1.

On 29 June, the NPF became noticeable at 08:35, causing the total $N_{cn}$ to increase rapidly by over 1 order of magnitude within 2 hours with a $FR$ of 2.3 cm$^{-3}$ s$^{-1}$ and $NMINP$ of $2.0 \times 10^4$ cm$^{-3}$. The newly formed particles took approximately 3 hours to grow from the initial median mode diameter of < 10 nm to the maximum median mode diameter of 20 nm with the corresponding half width of 12 nm in 99.7 % confidence. However, similar to our previous findings and other studies reviewed by Chu et al. (2019), new particles stopped growing after approximately 10 hours and could even slightly shrink before disappearing. With $\kappa$ values < 0.1, only atmospheric particles larger than 120–200 nm could be activated as CCN at 0.2 % and 0.4 % SS (Petters and Kreidensohler, 2007), which is conventionally referred to as the CCN-activated size. Even at 1.0 % SS, the CCN-activated size at $\kappa$ values < 0.1 (Fig. S9) should be larger than 70 nm. Therefore, the newly grown particles were too small to act as CCN, regardless of ambient SS, as previously reported (Hammer et al., 2014; Hudson et al., 2015; Shen et al., 2018). Moreover, the variations in $N_{ccn}$ at 0.2 % SS were likely determined by the number concentrations of larger pre-existing particles based on the correlation between $N_{ccn}$ at 0.2 % SS and $N_{cn>100}$ from 09:00 to 24:00 on 29 June. The regression equation can be expressed as follows: $N_{ccn} = N_{cn>100} \times 0.42 + 64$, with an R$^2$ of 0.70 and $p < 0.01$, at 0.2 % SS.

On 3 July, NPF event commenced at 08:10 with $FR$ of 0.75 cm$^{-3}$ s$^{-1}$ and $NMINP$ of $9.5 \times 10^3$ cm$^{-3}$ s$^{-1}$ (refer to Fig. 3d–f). However, between 06:00 to 13:00, the $N_{ccn}$ at the lab-calibrated 0.2 % and 0.4 % SS decreased with decreasing $\kappa$ values. Meanwhile, $N_{cn>100}$ slightly increased during the same time period. It is likely that less CCN-activated vapor condensation had a significant effect on the growth of pre-existing particles resulting in the increase of $N_{cn>100}$ and decrease of $N_{ccn}$. Following 13:00, $N_{ccn}$ values at 0.2 % and 0.4 % SS experienced two stepwise increases, accompanied by an increase in $N_{cn>100}$. For example, $N_{ccn}$ values at 0.2 % and 0.4 % SS almost doubled with the median mode diameter of the grown new particles narrowing to 25–33 nm from 13:00 to 15:00, and the two values then slightly decreased till 20:00. However, the grown new particles were still too small to be activated as CCN at 13:00–15:00 with the median mode diameter plus the corresponding half width of 24–30 nm in 99.7 % confidence to be considered. From 18:00 to 24:00, the maximum median mode diameter of the grown new particles stopped at 48 nm and the corresponding half width of 47 nm in 99.7 % confidence. The calculated $\kappa$ values were $0.13 \pm 0.03$ at 0.2 % SS and < 0.1 at 0.4 % SS ($0.15 \pm 0.03$ at 0.18 % SS and < 0.1 at 0.36 % SS). For CCN activation, particles need to be larger than 100–140 nm for lab-calibrated and approximated on-site SS. In this case, the increase in $N_{ccn}$ after 13:00 may have been due to the observed growth of pre-existing particles from ~ 50 nm to particles larger than 100 nm (refer to Fig. 3d). It is possible that the growth of pre-existing particles was driven by organic vapor with lower CCN activation since $\kappa$ values at 0.4 % SS decreased from 0.11 to lower values. However, the increasing size of the pre-

existing particles may have canceled out the decreasing effect of $\kappa$ on $N_{ccn}$. It is worth noting that even smaller $\kappa$ values were calculated at 1.0 % SS on 3 July (refer to Fig. S9). Thus, the grown new particles could not have contributed to $N_{ccn}$ at the lab-calibrated 1.0 % SS.

During the NPF events that occurred from 12–14 July (refer to Fig. S7a–c), no discernible increase in $N_{ccn}$ at the lab-calibrated 0.2 % and 0.4 % SS was observed in contrast to the events immediately prior. The three NPF events were often linked to the intrusion of various aerosol plumes. However, based on the combination of lower $\kappa$ values calculated at 0.2 % and 0.4 % SS and the size of the newly formed particles during these events, it can be inferred that there was likely no net contribution of the grown particles to the observed $N_{ccn}$.

### 3.3 Case study with positive contributions to $N_{ccn}$ during NPF events, but not from grown new particles

During NPF events on 30 June and 6 July, an increase in $N_{ccn}$ values at the lab-calibrated 0.2 % and 0.4 % SS was observed (see Fig. 4a–f). However, the maximum median mode diameter of the newly formed particles stopped at 19 ± 1 nm on 30 June and 25 ± 1 nm on 6 July with the corresponding half width of 28 nm and 37 nm in 99.7 % confidence, respectively (see Fig. 4a, d). On 30 June, the particles grew rapidly before 12:00, stopped growing, and even shrank slightly in the next 11 hours. On 6 July, the particles reached their maximum size at 13:00, stopped growing in the next 3 hours, and disappeared shortly after. The calculated $\kappa$ values at 0.2 % and 0.4 % SS were smaller than 0.1 on 30 June and the same was true at approximated on-site 0.18 % and 0.36 % SS. On 6 July, the maximum $\kappa$ values at 0.2 % SS were 0.21 ± 0.02 (or 0.26 ± 0.02 at 0.18 % SS) during 12:00–16:00, and the maximum value at 0.4 % was 0.16 (or 0.19 at 0.36 % SS) at 16:00. The small size of the newly formed particles made it unlikely for them to contribute to the increase in $N_{ccn}$ at 0.2 % and 0.4 % SS during the NPF events. Even at 1.0 % SS, the grown new particles were too small to act as CCN due to even smaller $\kappa$ values (see Fig. S9).

On 30 June, the observed increase in $N_{ccn}$ was partially attributed to the increased number concentration of pre-existing particles. This was demonstrated by a significant correlation between $N_{ccn}$ and $N_{cn>100}$ from 10:00 to 14:00 on that day, with an equation of $N_{ccn} = N_{cn>100} \times 1.42 - 5.6 \times 10^2$, $R^2 = 0.83$, and $p < 0.05$ at 0.2 % SS. However, on 6 July, there was no significant correlation between $N_{ccn}$ at 0.2 % SS and $N_{cn>100}$ during the NPF event, implying that the increase in $N_{ccn}$ at 0.2 % and 0.4 % SS was mainly due to the increased $\kappa$ values of the pre-existing particles (see Fig. 4e–f).

### 3.4 The contributions of grown new particles to $N_{ccn}$ at various SS on 1 July

In contrast to the seven NPF events discussed above, the newly formed particles experienced continuous growth from < 10 nm at 08:30 on 1 July to 65 ± 3 nm at 00:00–04:50 on 2 July (see Fig. 5a). The corresponding half width at the latter 5 hours varied from 153 nm to 203 nm with 99.7 % confidence and from 114 nm to 138 nm with 99 % confidence, respectively. Thus, at least 1 % of the grown new particles (61–73 cm$^{-3}$) were large enough to be activated as CCN. Supposed that the part of grown new particles were totally activated as cloud droplets, they should yield an appreciable contribution to $CNDC$. Local meteorological data recorded 0.2 mm rainfall at 04:00–05:00 on 2 July. However, after 04:50 on 2 July, the new particle signal was overwhelmed by the intrusion of aerosol plumes because the $N_{cn}$ varied

significantly with an invariant median accumulation mode diameter (see Fig. 2a). Spatial inhomogeneity of the NPF occurrence could not be completely excluded (Zhou et al., 2021). The *FR* of the NPF event was only 0.82 cm$^{-3}$ s$^{-1}$ on that day, suggesting a weak NPF event. This value ranked lower than the values of 0.06–5.95 cm$^{-3}$ s$^{-1}$ reported in other forest areas (Fiedler et al., 2005; Dal Maso et al., 2005; Han et al., 5    2013).

During the NPF event on 1 July, a large increase in $N_{ccn}$ was observed at 0.2 % SS (see Fig. 5b). Prior to the NPF event, the $N_{ccn}$ at 0.2 % SS had been $0.6 \pm 0.1 \times 10^3$ cm$^{-3}$ for 4 hours. During the event, there were three stepwise increases: the first stage was $0.8 \pm 0.06 \times 10^3$ cm$^{-3}$, followed by the second stage which reached $1.3 \pm 0.04 \times 10^3$ cm$^{-3}$ between 13:00 and 19:00. The $N_{ccn}$ then increased to a high 10    level and fluctuated around $1.8 \pm 0.2 \times 10^3$ cm$^{-3}$ (third stage) until the new particle signal disappeared at 04:00–05:00 on 2 July (see Fig. 5b). Overall, the $N_{ccn}$ at 0.2 % SS increased by approximately 200 % during the NPF event compared to the stable values prior to its occurrence.

The maximum $\kappa$ values at 0.2 % SS were calculated to be $0.30 \pm 0.03$ from 19:00 on 1 July to 05:00 on 2 July, which was close to the values observed before 06:00 on 29 June and during the non-NPF period 15    observed on 12–14 July. However, the calculated $\kappa$ values were below 0.2 before 19:00, with CCN-activated size larger than 120 nm. The new particles grew with median mode diameters below 43 nm plus half width below 12 nm in 99.7 % confidence, which were unlikely to contribute to $N_{ccn}$ before 19:00. The same was true when the approximated on-site 0.18 % SS was considered. Therefore, the increased $N_{ccn}$ were likely caused by the pre-existing particles with increasing $\kappa$ values. The hourly 20    average $N_{cn>100}$ stayed around $1.4 \pm 0.02 \times 10^3$ cm$^{-3}$ from 09:00 to 13:00 and $1.5 \pm 0.06 \times 10^3$ cm$^{-3}$ from 13:00 to 19:00, respectively. The stepwise increase in $N_{cn>100}$ was also likely to reflect the change in pre-existing particle number concentration instead of the continuous growth of newly formed particles and subsequent increase in $N_{cn>100}$.

From 19:00 on 1 July to 05:00 on 2 July, the $N_{ccn}$ at 0.2 % SS were highly correlated with the $N_{cn>100}$, 25    i.e., $N_{ccn} = N_{cn>100} \times 1.01 + 197$, R$^2$ = 0.89, $p$ < 0.01. However, the $N_{cn>100}$ varied around $1.6 \pm 0.2 \times 10^3$ cm$^{-3}$ during the period and exhibited a stable trend tested by the Mann-Kendall method with $p$ value of 0.44. The stable trend implied that the contribution from the grown new particles to $N_{cn>100}$ was statistically undetectable. The grown new particles were unlikely to compete with the pre-existing particles to form cloud droplets at 0.2 % SS because of their smaller sizes and lower $\kappa$ values, despite the 30    possibility of decreased ambient SS with rapid uptake of water vapor on particles (Crumeyrolle et al., 2021; Gong et al., 2023).

Upon analyzing the data at 0.4 % SS on 1–2 July, it was found that the newly grown particles likely contributed significantly to the observed increase in $N_{ccn}$ after 15:00 on 1 July. Prior to this time, although the $N_{ccn}$ at 0.4 % SS had increased, the calculated $\kappa$ values were below 0.1 for CCN-activated sizes larger 35    than 100 nm. However, the maximum median mode diameter of the newly grown particles was smaller than 27 nm with the corresponding half width of 12 nm in 99.7 % confidence before 15:00. Hence, the increase in $N_{ccn}$ was attributed to the increased $\kappa$ values of pre-existing particles. After 15:00, the increased $\kappa$ values led to a smaller CCN-activated size of 67–87 nm at 0.4 % SS. The larger difference in $N_{ccn}$ between 0.2 % and 0.4 % SS was likely due to the increased contribution of the newly grown 40    particles to $N_{ccn}$ at 0.4 % SS (refer to Fig. 5d). Note that the calculated $\kappa$ values at 0.4 % SS after 15:00

might suffer from an error to some extent (Wex et al., 2010). In this case, the grown new particles externally existed with pre-existing particles and were unable to satisfy the internally mixing assumption required in Eq. (1).

To quantify the contribution of grown new particles to $N_{ccn}$ at 0.4 % SS, we introduced a new term, $N_{ccn,diff}$, which represents the difference between $N_{ccn}$ at 0.4 % SS and $N_{ccn}$ at 0.2 % SS. We made two assumptions: first, that the $N_{ccn,diff}$ value at 14:00 (386 cm$^{-3}$) represented the $N_{ccn,diff}$ of pre-existing particles, and second, that the $N_{ccn,diff}$ values of pre-existing particles remained constant after 15:00. Therefore, the difference between $N_{ccn,diff}$ after 15:00 and $N_{ccn,diff}$ at 14:00 represented the net contribution of grown new particles to $N_{ccn}$ at 0.4 % SS. The net contribution of grown new particles was 316 ± 304 cm$^{-3}$ from 15:00 on 1 July to 05:00 on 2 July, accounting for only 12 ± 11 % of $N_{ccn}$ at 0.4 % SS. The maximum net contribution occurred at 16:00 on 1 July and was determined to be $1.0 \times 10^3$ cm$^{-3}$, accounting for 38 % of $N_{ccn}$ at 0.4 % SS. These rough estimates suggest that pre-existing particles were still the major contributor to $N_{ccn}$ at 0.4 % SS, outnumbering new particles.

We used the same method to analyze the contribution of grown new particles to $N_{ccn}$ at 1.0 % SS by calculating $N_{ccn,diff}$, which was the difference between $N_{ccn}$ at 1.0 % and 0.2 % SS (see Fig. 5e). However, using $N_{ccn,diff}$ at 11:00 appeared to be more reasonable than using $N_{ccn,diff}$ at 14:00 to estimate the net contribution of the grown new particles to $N_{ccn}$ at 1.0 % SS. The results of the test are presented below. We assumed the $N_{ccn,diff}$ value at 11:00 (864 cm$^{-3}$) to represent the $N_{ccn,diff}$ of pre-existing particles after 12:00, and assumed that the $N_{ccn,diff}$ was invariant after 12:00. It can obtain that the net contribution of the grown new particles was 769 ± 514 cm$^{-3}$ from 12:00 on 1 July to 05:00 on 2 July, accounting for only 23 ± 12 % of $N_{ccn}$ at 1.0 % SS. The maximum net contribution was $1.9 \times 10^3$ cm$^{-3}$ at 18:00 on 1 July, which accounted for 42 % of $N_{ccn}$ at 1.0 % SS. We also observed a minimum contribution of 4 % at 03:00 on 2 July, which was consistent with the disappearance of new particle signals. Alternatively, We attempted to use the $N_{ccn,diff}$ at 14:00 (1533 cm$^{-3}$) represented the substrate constant $N_{ccn,diff}$ of the pre-existing particles after 15:00. However, negative net contributions of the grown new particles to $N_{ccn}$ at 1.0 % SS were observed after 22:00 on 1 July, suggesting that the $N_{ccn,diff}$ of pre-existing particles was overestimated. The above-mentioned conclusions were also valid when the approximated on-site SS were used.

In the literature, the impact of NPF events on CCN has been widely investigated, e.g., the observational results from 35 sites worldwide summarized by Ren et al. (2021), and the studies can be classified into three categories. Category 1: the concentrations of CCN were measured and the contribution of grown new particles to CCN loadings at various SS levels were calculated by tentatively deducting the perturbation from varying pre-existing particles during the NPF events. Category 2: the concentrations of CCN were also measured during the NPF events. However, the pre-existing particles were assumed to be invariant when calculating the contribution of grown new particles to CCN loading during the NPF events. No tentative approach was conducted to deduct the perturbation from varying pre-existing particles on the increase in $N_{ccn}$ relative to those immediately observed before NPF events. Category 3: the concentrations of CCN were not directly measured. Instead, the values were estimated either by the size of grown new particles beyond a certain threshold or a combination of the grown new particle size and $\kappa$ values. Here, we compared our results with those previously reported at mountain

sites or several remote continental sites on this issue (Table 1).

For Category 1, there were only two studies reported in the literature. Cai et al. (2021) analyzed the contribution of 3–80 nm particles to $N_{ccn}$ at the Wudang Mountain during three NPF events when there were more than 50 %, 275 %, and 140 % increase in $N_{ccn}$ at 0.8 % SS. During the three events, the $D_p$ of grown new particles were below 40 nm (see Fig. 2a in Cai et al., 2021). However, their Fig. 2a also showed that the concentrations of pre-existing particles at 40–80 nm during the NPF events clearly increased relative to those observed immediately before the events. It seemed that the selected contribution of 3–80 nm particles to $N_{ccn}$ alone might be insufficient to deduct the perturbation from varying pre-existing particles. Moreover, Cai et al. (2021) didn't analyze the contribution at 0.2 % SS because the grown new particles were too small to be activated as CCN at 0.2 % SS. Rejano et al. (2021) compared the maximum increase during NPF events with that during non-NPF events, with a relative increase of 115 % at 0.25 % SS and 175 % at 0.5 % SS, respectively. In their study, they clearly assumed that the pre-existing particles were invariant between NPF events and non-NPF events. However, the assumption appeared to be invalid on basis of the results shown in their Fig. 4d and Fig. 6d, i.e., $N_{ccn}$ at 0.5 % SS peaked at 13:00 and then decreased. However, the $D_p$ of grown new particles at approximately 25 nm at 13:00 (shown in their Fig. 4b) were obviously too small to be activated as CCN at 0.5 % SS and the larger increase in $N_{ccn}$ before 13:00 might had a high possibility mainly due to varying pre-existing particles rather than grown new particles.

For Category 2, more studies were available in the literature. For example, Kim et al. (2019) reported $N_{ccn}$ at $0.2 \times 10^3$ cm$^{-3}$ at 0.4 % SS during the NPF events in the polar measurement and estimated the contribution of NPF to CCN at 11 %. Hirshorn et al. (2022) observed a substantial increase in $N_{ccn}$ at 0.2–0.4 % SS during the NPF events relative to those during the non-NPF days in winter (36 %) and spring (54 %) at a mountaintop site in America, while there were no significant increases in summer and fall. In a forest study conducted by Sihto et al. (2011), the $N_{ccn}$ at 0.2–0.6 % SS on NPF events was lower than those on non-NPF days. However, the $N_{ccn}$ during NPF events exhibited 70 %–110 % increases at 0.1–1.0 % SS relative to those observed immediately before NPF events. Other studies have also reported an increase in $N_{ccn}$ during the NPF events relative to those values immediately observed before, but they didn't report quantitative contributions (Creamean et al., 2011, Kawana et al., 2017, Pierce et al., 2012).

For Category 3, Rose et al. (2017) used 50 nm or 100 nm as lower limits for critical diameter ($D_c$) to activate as CCN and reported enhancements ranged from 250 % to 100 % for $CCN_{50}$ to $CCN_{100}$ during NPF events compared to non-NPF days. Same method was employed by Laakso et al. (2013) in Savannah, where more than 40 % improvement for $CCN_{60}$ was found during NPF events in wet season, but no significant improvement in dry season. Kalkavouras et al. (2019) employed particles size distribution and chemical composition measurements to determine the $D_c$, and found $N_{ccn}$ improvements ranged from 29 % to 77 % at SS levels of 0.1 % to 1.0 %. In addition to the three studies, there are dozens of studies including our previous one conducted at Tai Mountain (Zhu et al., 2021b) subject to Category 3. Due to lack of direct measurements of $N_{ccn}$, their uncertainties cannot be evaluated.

Overall, it appears that a big challenge still exists to reasonably deduct the perturbation from varying pre-existing particles in calculating the net contribution of $N_{ccn}$ from grown new particles in research community.

### 3.5 Hydrophilic organics dominated the new particle growth, but only NH₄NO₃ formation or hygroscopic organic condensation increased $N_{ccn}$

Organic species have been widely reported to participate in nucleation and play a significant role in driving the growth of newly formed particles, particularly in forested areas (Makela et al., 2001; Ehn et al., 2007; Smith et al., 2008, 2010; Riipinen et al., 2009). In our study, at 1.0 % SS, the calculated $\kappa$ values were consistently smaller than 0.13 during all NPF events, with a maximum $\kappa$ value of 0.08 $\pm$ 0.02 from 12:00 on 1 July to 05:00 on 2 July (see Fig. 5c). These results strongly suggest that less hygroscopic organics with low volatility were dominant in the growth of newly formed particles to large sizes. However, the calculated $\kappa$ values might suffer from the error to some extent on 1–2 July when grew new particles can be activated as CCN at 1.0 % SS. In the case, the external mixing of grew new particles and pre-existing particles cannot satisfy the assumption required by Eq. (1).

Sulfuric acid vapor has been widely acknowledged as a crucial factor in the growth of newly formed particles to the size required to become CCN (Birmili et al., 2003; Kulmala et al., 2004; Young et al., 2008; Boy et al., 2005; Kerminen et al., 2018; Chu et al., 2019; Cai et al., 2021). Bzdek et al. (2012) found that sulfate contributed to 29 %–46 % of the total mass in the grown new particles during two campaigns conducted in Delaware, USA. If $H_2SO_4$ condensation had surpassed organic condensation in shaping pre-existing particles and increasing their $\kappa$ values on 1 July and 6 July, its amount should have been sufficient to dominate newly formed particles. However, this scenario was practically impossible because the $\kappa$ values at 1.0 % SS were much lower than those at 0.2 % SS. Nonetheless, $\kappa$ values did increase during the NPF events on 1 July and 6 July at 0.2 % SS, and were significantly larger than those on other NPF days with $p < 0.05$. Although secondary formation of $H_2SO_4$ and its ammoniated salts on pre-existing particles can occur (Bzdek et al., 2012; Cai et al., 2021), the process was unlikely to be significant in growing newly formed particles based on the low $\kappa$ values at SS = 1.0 %. In fact, the concentrations of $SO_4^{2-}$ in TSP ranged from 0.9 to 4.9 µg m⁻³, and were not significantly increased on 1 July and 6 July when compared with those on the other six NPF days (see Table 2). These complex results indicate the importance of accurately measuring $SO_4^{2-}$ in different-sized nanometer particles.

NH₄NO₃ formation on newly formed particles was unlikely to occur on those 2 days based on the calculated low $\kappa$ values at 1.0 % SS. Theoretically, NH₄NO₃ formation requires the product of NH₃ and HNO₃ mixing ratios to be larger than its equilibrium constant plus the Kelvin effect term (Lee et al., 2019; Wang et al., 2020). However, on 1 July and 6 July, NH₄NO₃ formation may have taken over low CCN-activated organic condensation on pre-existing particles larger than 100 nm. For example, the concentrations of $NO_3^-$ were 1.0 µg m⁻³ and 1.8 µg m⁻³ on 1 July and 6 July, respectively, and these values were significantly higher than those on the other six NPF days (0.5–1.0 µg m⁻³) with $p < 0.05$. Additionally, the SOC concentrations of 1.8–2.3 µg m⁻³ on 1 July and 6 July were significantly lower than those of 2.7–4.0 µg m⁻³ on the other 6 days with $p < 0.05$. The organic condensation on newly formed and pre-existing particles was likely reduced, indirectly enhancing the NH₄NO₃ formation effects on increasing $N_{ccn}$ and $\kappa$ values. However, the size-dependent chemical composition of atmospheric particles needs to be confirmed for further analysis.

We further examined four types of secondary organic tracers derived from isoprene, monoterpene, sesquiterpene, and aromatics, as well as primary organic tracers including levoglucosan (LEVO),

mannosan, and galactosan, between the two NPF periods (see Table 2 and Table S1). However, there was no significant difference in the concentrations of any type of organic tracers between the two NPF periods. It is worth noting that oxalic acid and its salts had high $\kappa$ values and could be important contributors to increasing $\kappa$ values at various SS on 1 July and 6 July, particularly during the daytime. However, the

measured concentrations of oxalate on those 2 days did not show a significant increase compared to the other days. In the literature (Rollins et al., 2012; Ehn et al., 2014), organonitrates were reported as an important secondary aerosol composition at nighttime. However, the species were not measured in this study. Thus, the influence of organonitrates on $\kappa$ values of the observed atmospheric particles is unknown.

We examined the satellite-derived $CDNC$ over the mountain area during and after the NPF days.

The values were 169 cm$^{-3}$, 89 cm$^{-3}$, and 101 cm$^{-3}$ on 1, 2, and 3 July, respectively (see Fig. 2d). These values were approximately 1 order of magnitude smaller than the observed $N_{ccn}$ at 0.2 % SS on those 3 days. The $CDNC$ is a strong function of the actual SS present in the atmosphere in addition to $N_{ccn}$ and the actual SS is determined by the ascent velocity of the air mass, the amount of moisture, etc (Pruppacher and Klett, 2012). This large difference between the observed $N_{ccn}$ and satellite-derived

$CDNC$ implies that the actual SS in the atmosphere might be substantially smaller than 0.2 %. In fact, Gao et al. (2021) recently conducted aircraft observations over Beijing and calculated the SS at cloud base to be approximately 0.048 %. Moreover, Shen et al. (2018) also reported the actual SS values varying from 0.01 % to 0.05 % during fog events observed in the NCP. Iwamoto et al. (2021) reported the mean SS around 0.34 % during cloud-shrouded periods at Mt. Fuji in Japan. Notably, their observed

SS decreased to 0.24 % when the air mass originated from continental sources. The higher SS observed at Mt. Fuji might be related to substantially lower $N_{ccn}$ (around 108 cm$^{-3}$ at 0.21 % SS) than those in Beijing. The reduced effect on SS levels with increasing $N_{ccn}$ was also obtained in Oklahoma (Jia et al., 2019), in which the estimated SS of stratocumulus and cumulus in relatively polluted atmospheres approximately equaled to 0.2 %. However, the $N_{ccn}$ in their relatively polluted atmospheres were smaller

than half of the observed $N_{ccn}$ in this study. Thus, it is not surprising to find that only a small fraction of CCN could competitively capture water vapor to form cloud droplets during the study period (Shen et al., 2018; Jiang et al., 2021; Gong et al., 2023). Moreover, the $CDNC$ during the period from 29 June to 14 July was 138 ± 99 cm$^{-3}$ and close to the seasonal average in June and July. The satellite-derived $CDNC$ on 2–3 July were even lower than the average, suggesting that the NPF event was unlikely to have any

influence on $CNDC$ at such low actual SS.

When the $COT$ and $CER$ were compared during NPF days and non-NPF days, the former $COT$ values around 14.2 ± 5.9 had no significant difference from the latter values around 17.2 ± 11.8 with $p$ = 0.53. The same was true for the $CER$ values, i.e., 13.2 ± 3.1 μm during NPF days versus 16.8 ± 4.0 μm during non-NPF days with $p$ = 0.067. Lack of significant differences on the two cloud parameters

between NPF days and non-NPF days might be related to the small dataset.

**3.6 Molecular evidence for organics dominating the new particle growth**

Figures 6a, b, and S10–S12 compare the static SIMS spectra of atmospheric nanometer particles collected on 30 June and 1 July. On 30 June, the collected particles with diameters of 60 nm (09:20–11:20), 100 nm (07:00–09:00), and 200 nm (13:30–15:30) likely represented the pre-existing particles in

the atmosphere, since the maximum median mode diameter of the grown new particles was only $19 \pm 1$ nm. In the lower mass range of $m/z^+$ 0–200, the fragments of organics suffered from strong interference from the substrate material (Fig. S10) and were not included in the analysis. The same was true on 1 July. However, the interference from the substrate in the mass range of $m/z^+$ 200–350 was negligible and was analyzed. Large differences in organic fragment peaks were observed in the mass region of $m/z^+$ 200–350 between nanometer particles collected on 30 June and 1 July (Fig. 6a, b). Organic peaks, such as $m/z^+$ 207.047 $C_{14}H_7O_2^+$, 221.158 $C_{14}H_{21}O_2^+$, 265.053 $C_{16}H_9O_4^+$, 267.059 $C_{16}H_{11}O_4^+$, 281.081 $C_{17}H_{13}O_4^+$, 325.013 $C_6H_{13}O_{16}^+$, and 327.018 $C_6H_{15}O_{16}^+$, appeared on particle surfaces with sizes of 60 nm, 100 nm, and 200 nm on 30 June. These organic fragment peaks rarely appeared or had much lower intensities on particle surfaces with sizes of 60 nm (13:00–15:00), 100 nm (18:23–20:23), and 200 nm (06:10–08:10) on 1 July. However, the organic fragment peaks such as $m/z^+$ $C_{14}H_7O_2^+$, $C_{14}H_{21}O_2^+$, $C_{17}H_{13}O_4^+$, and $C_6H_{15}O_{16}^+$ were detected with high intensities on the surface of 30 nm particles collected at 15:10–17:10 on 1 July. The collected particles with a diameter of 30 nm mainly consisted of grown new particles as shown in Fig. 5a. The results suggest that high-molecular-weight organic vapors may preferentially condense on the nanometer particles. However, this was not the case for particles with sizes of 60 nm, 100 nm, and 200 nm on 1 July, when semi-volatile organic and inorganic vapors may have overwhelmingly condensed on the sized particle surfaces and covered up the high-molecular-weight organic fragment signals.

The ToF-SIMS spectral comparisons in the negative ion mode (Figs. S11–S12) yielded similar results, which support the conclusion that the high-molecular-weight organic fragment signals were concealed on the surfaces of the sampled nanometer particles with diameters of 60 nm, 100 nm, and 200 nm on 1 July.

The selected peak spectral of SIMS data were also analyzed to identify variations among the samples using Principal Component Analysis (PCA) (Fig. 7a–c). Scores plots were generated to show the similarities and dissimilarities among the samples. The most important principal component (PC1) and the second important principal component (PC2) explained 84.5 % and 9.1 % of the data, respectively. The former separated the 30 nm particles collected on 1 July from the 100 nm and 200 nm particles collected on 30 June, as well as from the 60 nm particles collected on 30 June and 1 July (Fig. 7a). This indicates that PC1 positive loadings shared commonalities for the 30 nm particles on 1 July, and the 100 nm and 200 nm particles on 30 June. The organic fragment signals of the 30 nm particles collected on 1 July were significantly different from those of the larger particles on the same day. Some characteristic peaks were identified as components of organics, such as $m/z^+$ 73 $C_4H_9O^+$, 131 $C_8H_5NO^+$, 133 $C_4H_9N_2O_3^+$, 147 $C_6H_{15}SN_2^+$, 161 $C_9H_{21}O_2^+$, 207 $C_{14}H_7O_2^+$, 221 $C_{14}H_{21}O_2^+$, and 281 $C_{17}H_{13}O_4^+$ (Fig. 7b), and they contributed to PC1 positive loadings. These findings support the conclusion that organic vapors drove the condensation growth of 30 nm particles on 30 June and the sized particles on 1 July. Additionally, the PCA results in the negative ion mode showed consistent results with those in positive ion mode (Fig. S13).

**4 Summary**

During the 2 weeks field campaign from 29 June to 14 July 2019 at a rural mountain site in NCP, we observed and analyzed eight NPF events. On these NPF days, the total $N_{cn}$ was $8.4 \pm 6.1 \times 10^3$ cm$^{-3}$, which was substantially higher than the $4.7 \pm 2.6 \times 10^3$ cm$^{-3}$ observed on non-NPF days. However, the $N_{ccn}$ at 0.2 % SS and 0.4 % SS on NPF days was significantly lower than that on non-NPF days, with a $p$ value less than 0.05. For instance, the $N_{ccn}$ at 0.2 % SS was $1.2 \pm 0.7 \times 10^3$ cm$^{-3}$ on NPF days versus $1.6 \pm 0.8 \times 10^3$ cm$^{-3}$ on non-NPF days. Although the observational data size in this study is small, the comparison suggests that NPF events may not statistically increase $N_{ccn}$ in the middle and low SS levels. In 5 of the 8 NPF events, we observed either decreases or irregular changes in $N_{ccn}$ with decreasing $\kappa$ values at various SS. The decreasing $\kappa$ values were likely attributed to the condensation of hydrophilic organics on newly formed particles and pre-existing particles. Only in 1 of the 8 NPF events did the grown new particles yield detectable net contributions to $N_{ccn}$ at 0.4 % SS and 1.0 % SS with larger sizes of grown new particles and increased $\kappa$ values. The detectable net contributions accounted for $12 \pm 11$ % at 0.4 % SS and $23 \pm 12$ % of $N_{ccn}$ at 1.0 % SS, respectively, during the latter growth period of the NPF event. The above-mentioned conclusions were also valid when the approximated on-site SS at 0.18 %, 0.36 % and 0.9 % were used.

During all eight NPF events, the estimated $\kappa$ values at 1.0 % SS were below 0.13 (or 0.16 at 0.9 % SS), indicating that hydrophilic organics played a crucial role in the growth of newly formed particles. The estimated $\kappa$ values at 0.2 % SS were generally smaller than 0.20 (or 0.25 at 0.18 % SS) and decreased during 6 of the 8 NPF events. In these cases, less hygroscopic organic vapors likely condensed on pre-existing particles, reducing their CCN activation. However, on two of the NPF events (1 July and 6 July), the $\kappa$ values at 0.2 % SS increased. The significantly higher concentrations of $NO_3^-$ on those days suggested that $NH_4NO_3$ formation may have contributed to the increased $\kappa$ values at 0.2 % SS. However, $NH_4NO_3$ formation on grown new particles was unlikely due to the Kelvin effect.

On 1 July, the SIMS results indicated that high-molecular-weight organic vapors preferentially condensed on the 30 nm particles. In contrast, on > 60 nm particle surfaces, inorganic vapors overwhelmingly condensed, concealing the high-molecular-weight organic fragment signals. However, these signals were consistently detected on > 60 nm particle surfaces on 30 June due to limited or no condensation of inorganic vapors.

The contribution of NPF events to $N_{ccn}$ needs to be re-evaluated by taking into account the condensation of organic vapors or the formation of $NH_4NO_3$ on pre-existing particles. Moreover, the $N_{ccn}$ observed in the rural mountain atmosphere was considerably higher than the cloud droplet number concentrations derived from satellites. This suggests that the actual SS required to form cloud droplets in the NCP atmospheres may be substantially smaller than 0.2 %. Again, the grown new particles didn't yield a detectable contribution to $N_{ccn}$ at 0.2 % SS during all eight NPF events. Thus, it is reasonably argued that the grown new particles might not act as $N_{ccn}$ and form cloud droplets in NCP atmospheres with actual SS largely smaller than 0.2 %.

**Data availability.** The data of this paper are available upon contact with the authors, Xiaohong Yao (xhyao@ouc.edu.cn) and Xing Wei (17667565924@163.com). The MODIS level-3 daily averaged product (MYD08_D3) used in this paper are available from the Level-3 and Atmosphere Archive and

Distribution System (LAADS) Distributed Active Archive Center (DAAC) of NASA (http://ladsweb. nascom.nasa.gov/).

**Author contributions.** XY (Xiaohong Yao) designed the experiments. XYY (Xiao-Ying Yu) supervised the SIMS analysis. XW and YS analyzed the data and wrote the paper. XY, XYY, HG and YG provided advice on data processing. XY and XYY revised the original draft of the paper. All authors contributed to editing and improving the paper.

**Competing interests.** The authors declare that they have no conflict of interest.

**Acknowledgement.** This research was supported by the National Natural Science Foundation of China (No. 42276036) and Hainan Provincial Natural Science Foundation of China (No. 422MS098). The manuscript preparation for XYY was supported partially by the strategic Laboratory Directed Research and Development (LDRD) of the Physical Sciences Directorate of the Oak Ridge National Laboratory (ORNL).

The authors acknowledge the Beijing Forest Experimental Station of the Institute of Botany, Chinese Academy of Sciences for the help of logistics and permission to access to the site. ChatGPT was used to polish the language paragraph-by-paragraph.

This manuscript has been authored by UT-Battelle, LLC under Contract No. DE-AC05-00OR22725 with the U.S. Department of Energy. The United States Government retains and the publisher, by accepting the article for publication, acknowledges that the United States Government retains a non-exclusive, paid-up, irrevocable, world-wide license to publish or reproduce the published form of this manuscript, or allow others to do so, for United States Government purposes. The Department of Energy will provide public access to these results of federally sponsored research in accordance with the DOE Public Access Plan (http://energy.gov/downloads/doe-public-access-plan).

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

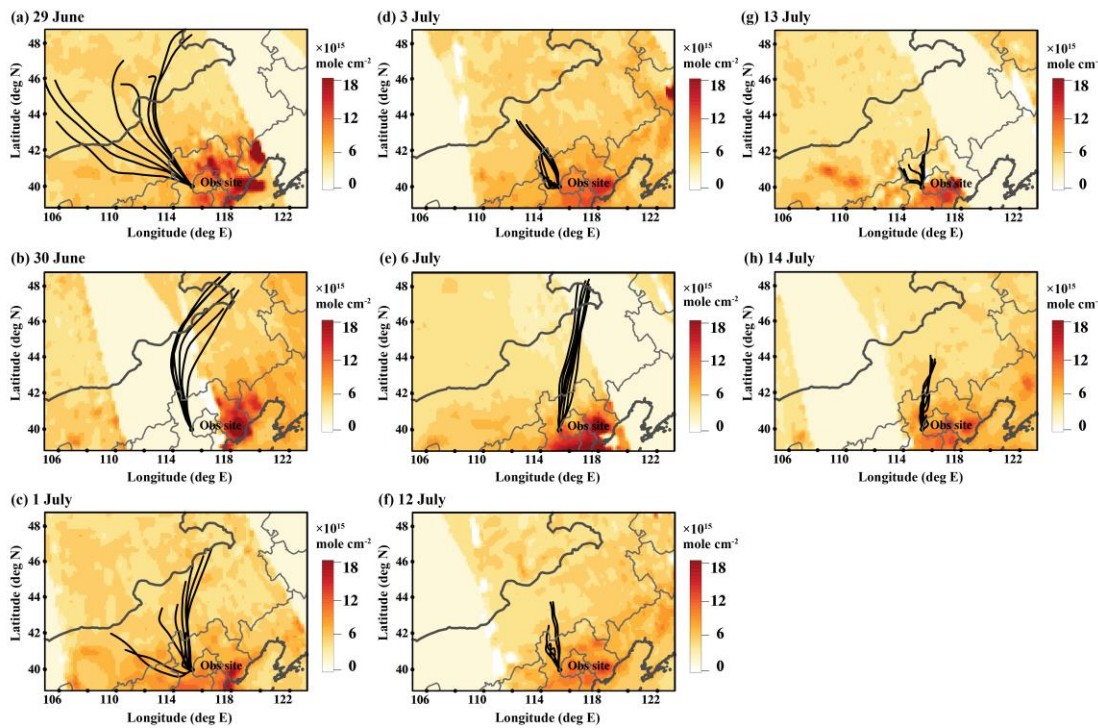

**Figure 1. Satellite NO₂ column density over the observational site and surrounding areas and 24-hour air mass back trajectories during eight NPF event days in June–July 2019 (a–f corresponded to the NO₂ column density recorded on 29 June, 30 June, 1 July, 3 July, 6 July, 12–14 July, respectively).**

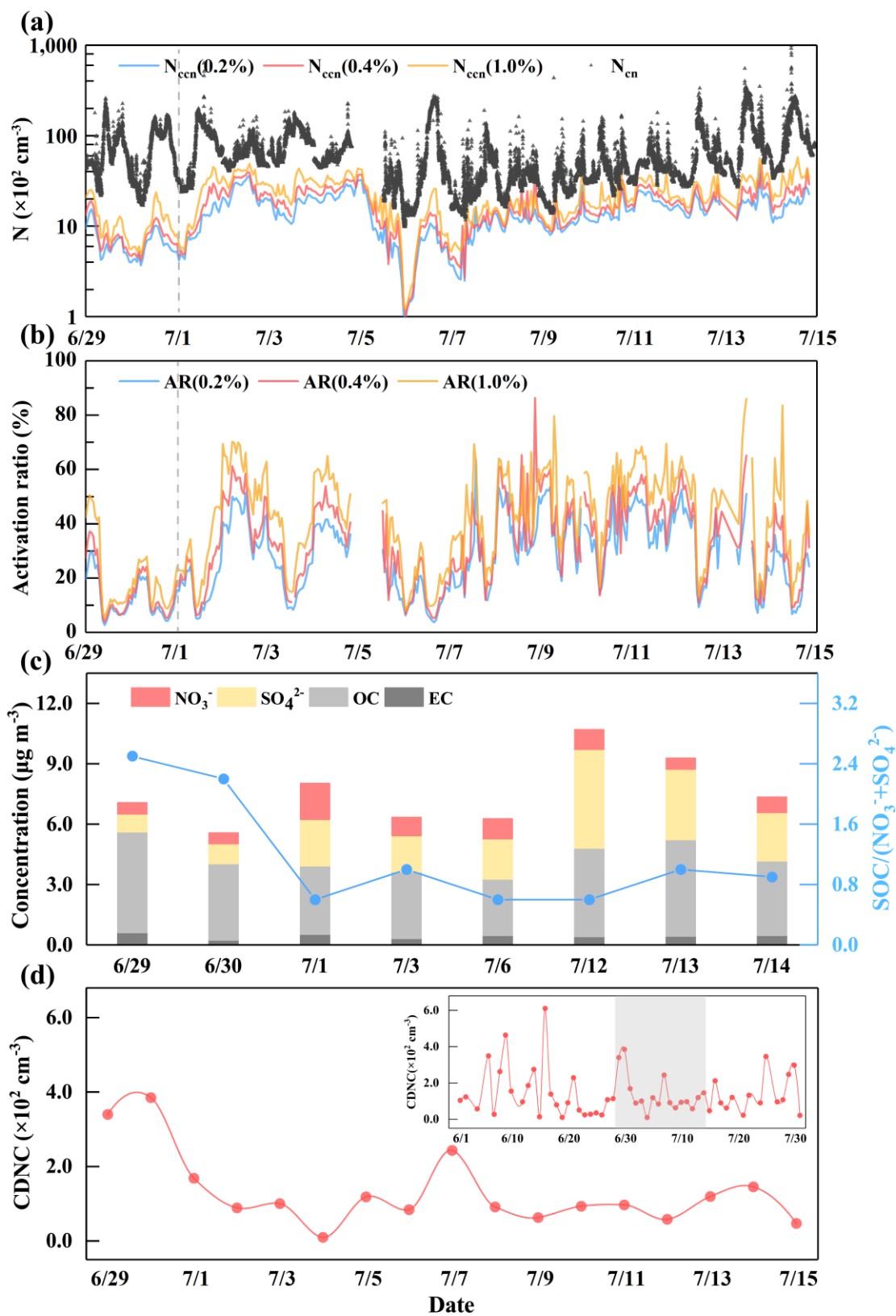

**Figure 2. Temporal variations in $N_{ccn}$ at 0.2 %, 0.4 % and 1.0 % SS, $N_{cn}$ (a), $AR$ at 0.2 %, 0.4 % and 1.0 % SS (b); Daily average concentrations of chemical components in TSP and related ratios of SOC/(NO$_3^-$ + SO$_4^{2-}$) (c); Satellite-derived $CDNC$ from 29 June 2019 to 15 July 2019 ($CNDC$ in June to July were superimposed in d) (d).**

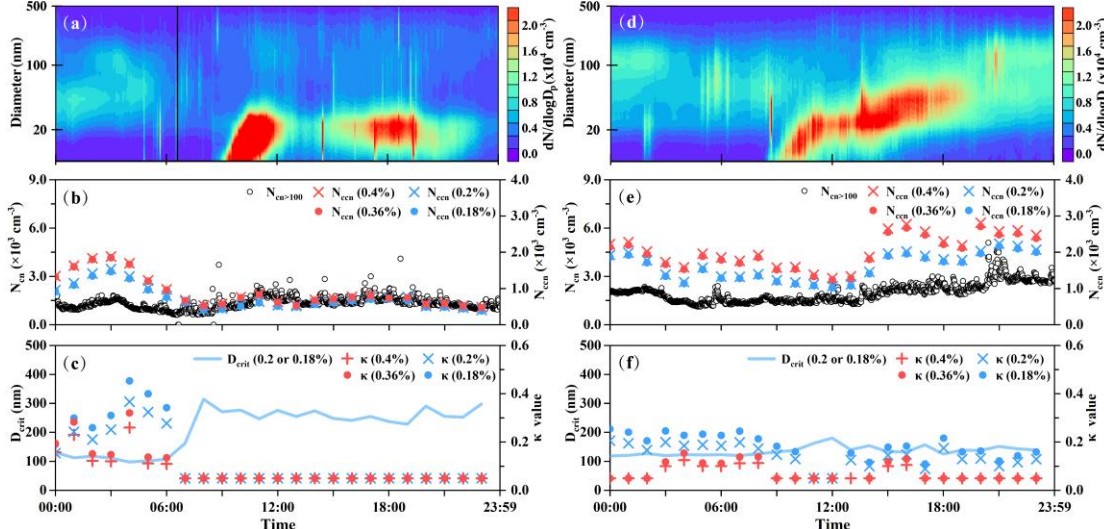

**Figure 3. The contour plots of particle number size distribution, time series for $N_{cn>100}$, and time series for $N_{ccn}$, $\kappa$ values and $D_{crit}$ at the lab-calibrated SS and the corresponding on-site approximated values, on 29 June and 3 July, respectively (a and d: Contour plots of *PNSD*; b and e: $N_{cn>100}$, $N_{ccn}$ at the lab-calibrated 0.2%, 0.4% SS, and the on-site approximated $N_{ccn}$ at 0.18%, 0.36% SS; c and f: $D_{crit}$ at 0.2% or 0.18% SS, $\kappa$ values at the lab-calibrated 0.2%, 0.4% SS, and the on-site approximated $\kappa$ values at 0.18%, 0.36% SS).**

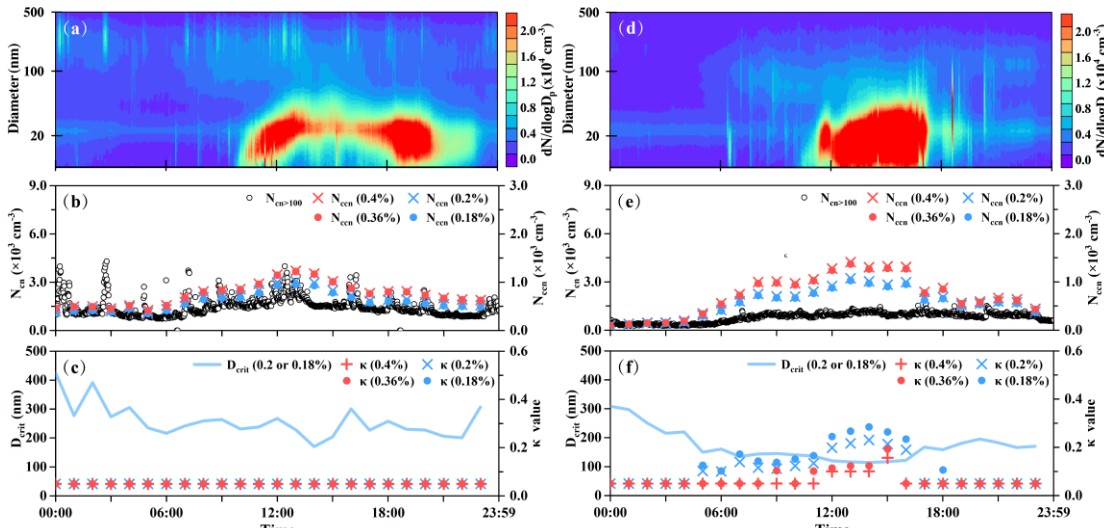

**Figure 4. The contour plots of particle number size distribution, time series for $N_{cn>100}$, and time series for $N_{ccn}$, κ values and $D_{crit}$ at the lab-calibrated SS and the corresponding on-site approximated values, on 30 June and 6 July, respectively (a and d: Contour plots of *PNSD*; b and e: $N_{cn>100}$, $N_{ccn}$ at the lab-calibrated 0.2%, 0.4% SS, and the on-site approximated $N_{ccn}$ at 0.18%, 0.36% SS; c and f: $D_{crit}$ at 0.2% or 0.18% SS, κ values at the lab-calibrated 0.2%, 0.4% SS, and the on-site approximated κ values at 0.18%, 0.36% SS).**

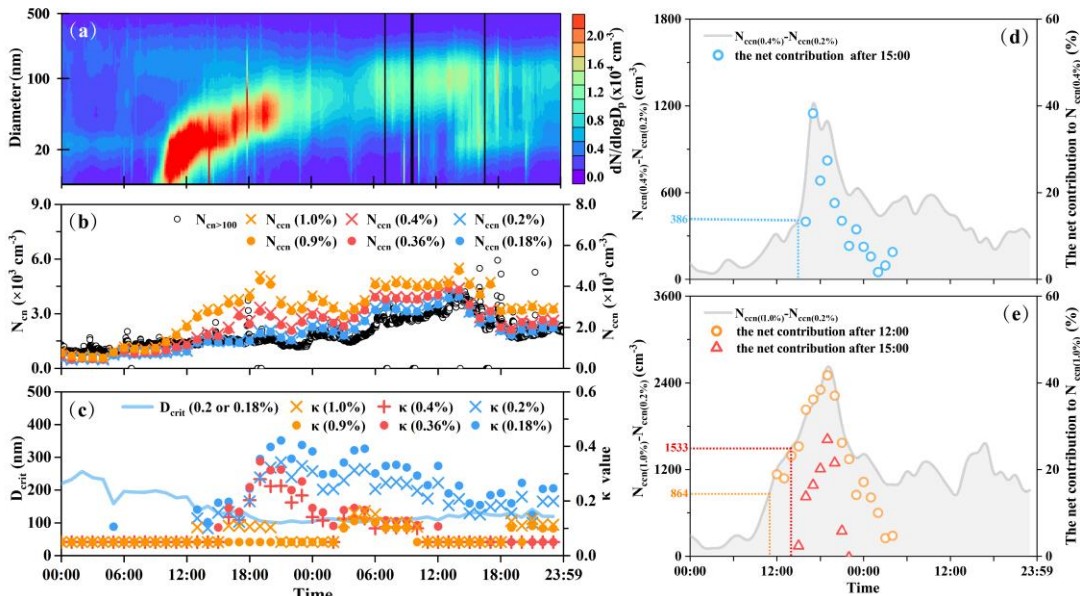

**Figure 5. The contour plots of particle number size distribution (a); Time series for $N_{cn>100}$, $N_{ccn}$ at the lab-calibrated 0.2%, 0.4% SS, and the on-site approximated $N_{ccn}$ at 0.18%, 0.36% SS (b); Time series for $D_{crit}$ at 0.2% or 0.18% SS, $\kappa$ values at the lab-calibrated 0.2%, 0.4% SS, and the on-site approximated $\kappa$ values at 0.18%, 0.36% SS (c); Time series of $(N_{ccn(0.4\%)} - N_{ccn(0.2\%)})$ and ratios of $(N_{ccn(0.4\%)} - N_{ccn(0.2\%)})/N_{ccn(0.4\%)}$ (d); Time series of $(N_{ccn(1.0\%)} - N_{ccn(0.2\%)})$ ratios of $(N_{ccn(1.0\%)} - N_{ccn(0.2\%)})/N_{ccn(1.0\%)}$ (e) on 1–2 July.**

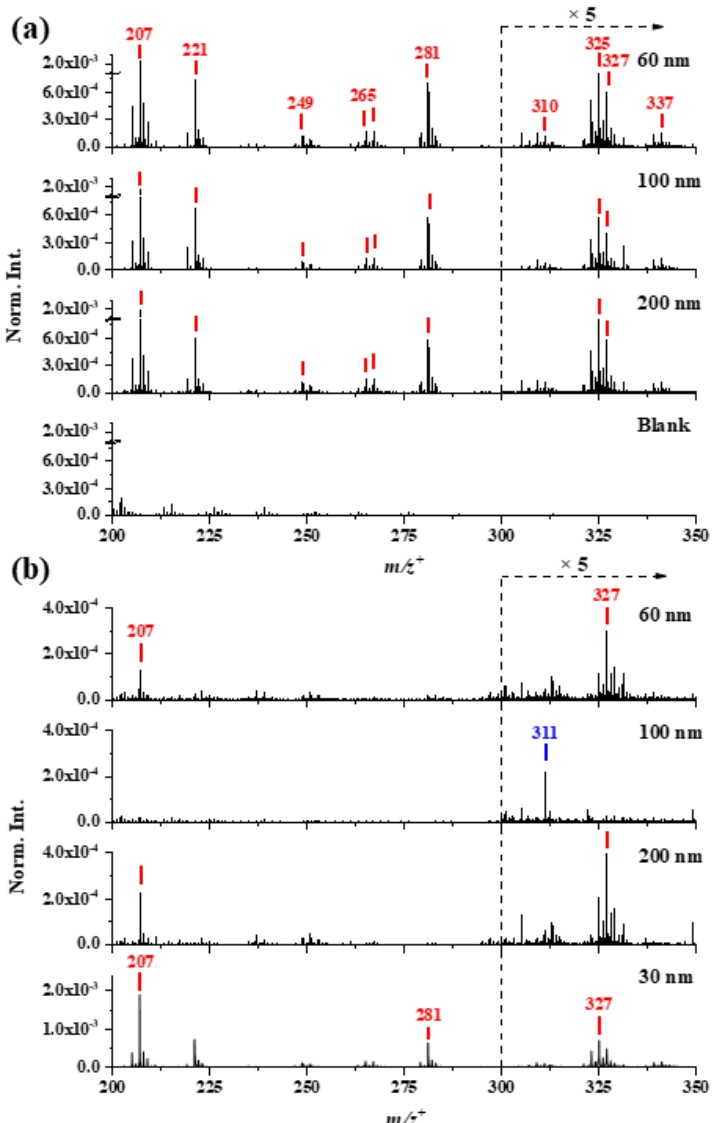

**Figure 6. ToF-SIMS spectral comparison of atmospheric nanometer particles collected on 30 June (a) and 1 July 2019 (b) in the positive ion mode ($m/z^+$ 200–350).**

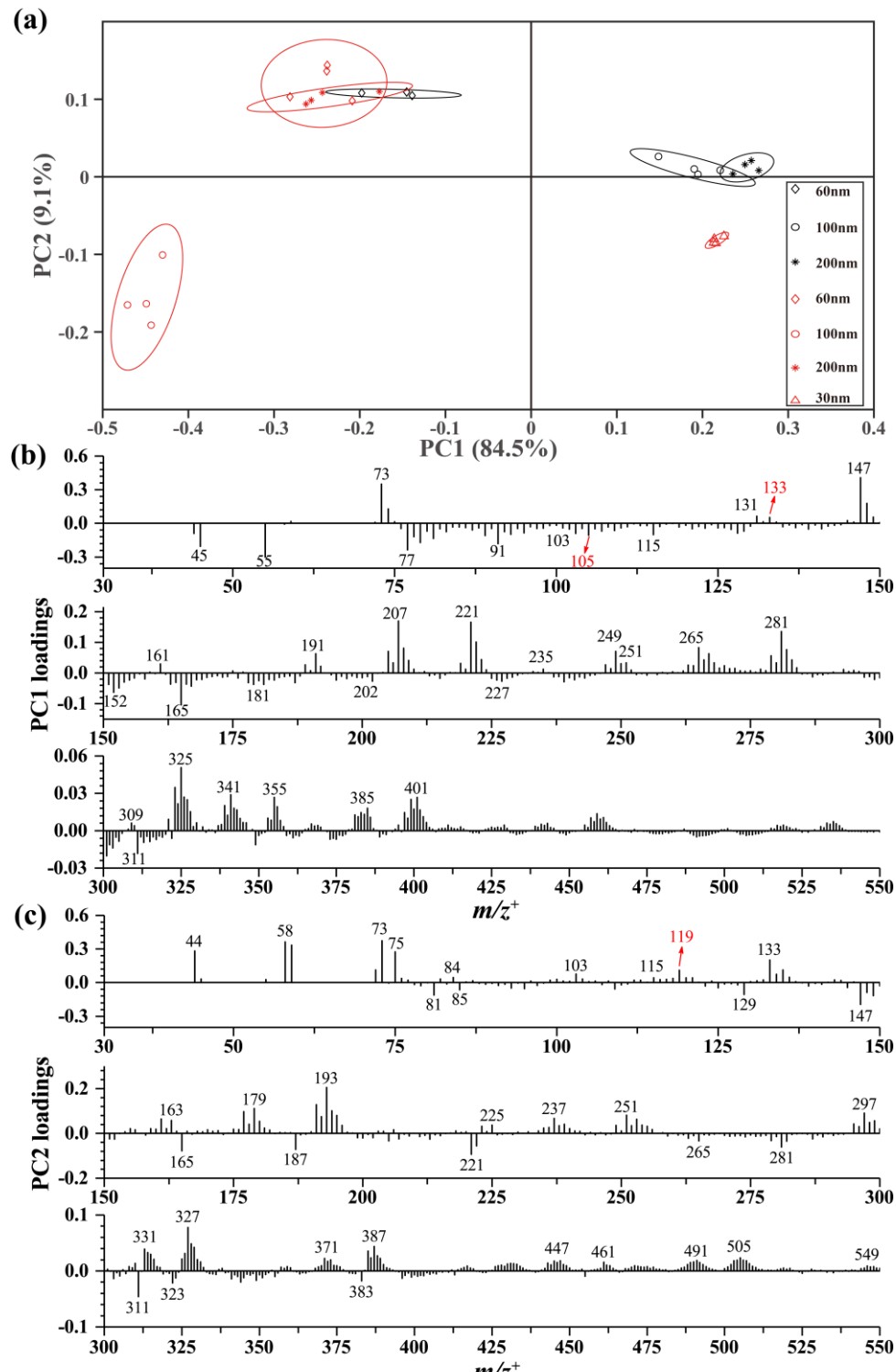

**Figure 7. ToF-SIMS selected peak spectral PCA results of 60 nm, 100 nm, and 200 nm particles on 30 June (gray markers) as well as 30 nm, 60 nm, 100 nm, and 200 nm particles on 1 July (red markers) in the positive mode: Scores plots of PC1 vs. PC2 (a), PC1 loadings plots in $m/z^+$ 30–550 (b), and PC2 loading plots in $m/z^+$ 30–550 (c). Peaks are labelled in their center masses.**

**Table 1. Details regarding the impact of NPF to $N_{ccn}$ at various sites. The information includes the percentage improvement of CCN attributable to NPF, as well as the number of NPF events that increase CCN.**

| Observation site | Reference [a] | Environment type | Observation date | NPF frequency [b] | $N_{ccn}$ on NPF days [c] | Impact of NPF to $N_{ccn}$ |
|---|---|---|---|---|---|---|
| Forest ecosystem research station of the Chinese Academy of Science, China | This study* | Mountaintop | June 2019 to July 2019 | 41 % (22) | $1.2 \pm 0.7 \times 10^3$ at 0.2 % SS; $1.5 \pm 0.9 \times 10^3$ at 0.4 % SS; $1.9 \pm 1.2 \times 10^3$ at 0.8 % SS | No significant improvement for $N_{ccn}$ at 0.2 % SS relative to non-NPF days |
| Sugar Pine Reservoir in Foresthill, America | Creamean et al. (2011) | Mountaintop | February 2009 to March 2009 | 33 % (18) | NA (no direct data source) | Newly formed particles can serve as $N_{ccn}$ at 0.3 % SS |
| Chacaltaya Global Atmospheric Watch station, Bolivia | Rose et al. (2017) * | Mountaintop | 2012 | 41 % (362) | NA (estimated by particle number size distribution) | 61 % of NPF events can contribute to $N_{ccn}$, more than 250 %, 150 %,100 % improvement for $CCN_{50}$, $CCN_{80}$ and $CCN_{100}$ relative to non-NPF days |
| Wudang Mountains air quality monitoring station, China | Cai et al. (2021) * | Mountaintop | May 2018 to June 2018 | 9 % (32) | $2.0 \pm 1.2 \times 10^3$ cm$^{-3}$ at 0.2 % SS; $4.2 \pm 2.8 \times 10^3$ cm$^{-3}$ at 0.8 % SS | 50 %, 275 %, 140 % improvement for $N_{ccn}$ at 0.8 % during the 3 NPF events |
| Sierra Nevada mountain alpine station, Spain | Rejano et al. (2021) * | Mountaintop | June 2019 to August 2019 | 70 % (96) | $0.8 \pm 0.6 \times 10^3$ cm$^{-3}$ at 0.5 % SS | 115 % improvement for $N_{ccn}$ at 0.25 % SS; 175 % improvement for $N_{ccn}$ at 0.5 % SS relative to non-NPF days |
| Storm Peak Laboratory, America | Hirshorn et al. (2022) * | Mountaintop | 2006 to 2021 | 50 % (835) | $0.2 \times 10^3$ cm$^{-3}$ between 0.2 % SS and 0.4 % SS | Winter: 36 % improvement for $N_{ccn}$ between 0.2 % SS and 0.4 % SS; Spring: 54 % improvement for $N_{ccn}$ between 0.2 % SS and 0.4 % SS; Summer and |

| | | | | | | fall: no significant improvement relative to non-NPF days |
|---|---|---|---|---|---|---|
| Whistler Mountain site, Canada | Pierce et al. (2012) * | Mountaintop | July 2010 | 83 % (6) | NA (no direct data source) | 3 of 5 NPF events can contribute to $N_{ccn}$ |
| Botsalano game reserve, South Africa | Laakso et al. (2013) | Savannah | July 2006 to February 2008 | 69 % (368) | NA (estimated by particle number size distribution) | Wet season: more than 40 % improvement for $CCN_{60}$ during NPF events Dry season: no significant improvement for $CCN_{60}$ during NPF events |
| King Sejong Station, Antarctic | Kim et al. (2019) | Polar | March 2009 to December 2012 | 6 % (1655) | $0.2 \times 10^3$ cm$^{-3}$ at 0.4 % SS | 11 % improvement for $N_{ccn}$ at 0.4 % SS during NPF events |
| Atmospheric observation station of the University of Crete, Greece | Kalkavouras et al. (2019) | Island | June 2008 to May 2015 | 162 NPF events (total observation days were not available) | NA (estimated by particle size distribution and chemical composition) | 29 %–77 % improvement for $N_{ccn}$ at 0.1–1.0 % SS during NPF events |
| Wakayama Forest Research Station, Japan | Kawana et al. (2017) | Forest | August 2010 | 40 % (10) | $0.6 \times 10^2$ cm$^{-3}$ at 0.12 % SS; $1.1 \times 10^2$ cm$^{-3}$ at 0.23 % SS; $1.5 \times 10^2$ cm$^{-3}$ at 0.41 % SS | all NPF events can contribute to $N_{ccn}$ at 0.12–0.41 % SS |
| SMEAR II station, Finland | Sihto et al. (2011) | Forest | July 2008 to June 2009 | 21 % (365) | $0.1 \times 10^3$–$0.2 \times 10^3$ cm$^{-3}$ at 0.1 % SS; $0.2 \times 10^3$–$0.4 \times 10^3$ cm$^{-3}$ at 0.2 % SS; $0.4 \times 10^3$–$0.8 \times 10^3$ cm$^{-3}$ at 0.4 % SS; $0.4 \times 10^3$–$0.9 \times 10^3$ cm$^{-3}$ at 0.6 % SS; $0.5 \times 10^3$–$1.3 \times 10^3$ cm$^{-3}$ at 1.0 % SS | 106 %, 110 %, 70 %, 82 %, 70 % improvement for $N_{ccn}$ at 0.1 %, 0.2 %, 0.4 %, 0.6 %, 1.0 % SS during NPF events |

a. Events that remove influence of pre-existing particles were indicated with asterisks.

b. The total observation days of data in the parentheses.

c. NA refer to the CCN observation data was not available, and the reasons are indicated in parentheses.

**Table 2. Concentrations of SOA tracers, OC, EC, and ions in TSP on NPF days**

| Chemical components | 29 June | 30 Jun | 1 July | 3 July | 6 July | 12 July | 13 July | 14 July |
|---|---|---|---|---|---|---|---|---|
| Isoprene SOA tracers (ng m$^{-3}$) | | | | | | | | |
| 2-methylglyceric acid | 0.04 | 0.44 | 0.65 | 1.9 | 0.21 | 0.17 | 6.2 | 1.4 |
| cis-2-Methyl-1,3,4-trihydroxy-1-butene | 0.12 | 1.0 | 0.59 | 1.2 | 0.98 | 2.5 | 36 | 39 |
| 3-Methyl-2,3,4-trihydroxy-1-butene | 0.24 | 1.61 | 0.50 | 0.52 | 0.31 | 1.10 | 16.39 | 23 |
| trans-2-Methyl-1,3,4-trihydroxy-1-butene | 0.15 | 0.73 | 0.36 | 0.02 | 0.16 | 1.2 | 17 | 13 |
| 2-methylthreitol | 1.2 | 11 | 5.8 | 2.3 | 5.2 | 0.72 | 32 | $1.0 \times 10^2$ |
| 2-methylerythritol | 2.9 | 22 | 17 | 7.6 | 15 | 1.3 | 75 | $2.2 \times 10^2$ |
| Sum of them | 4.6 | 37 | 25 | 14 | 22 | 7.0 | $1.8 \times 10^2$ | $4.0 \times 10^2$ |
| Biomass burning tracer (ng m$^{-3}$), OC and EC (μg m$^{-3}$) | | | | | | | | |
| levoglucosan | 2.6 | 11 | 0.21 | 2.2 | 0.44 | 0.16 | 0.84 | 0.72 |
| OC | 5.0 | 3.8 | 3.4 | 3.3 | 2.8 | 4.4 | 4.9 | 3.7 |
| EC | 0.61 | 0.23 | 0.53 | 0.32 | 0.47 | 0.41 | 0.43 | 0.47 |
| Water-soluble inorganic and organic ions (μg m$^{-3}$) | | | | | | | | |
| NO$_3^-$ | 0.57 | 0.54 | 1.8 | 0.92 | 1.0 | 0.99 | 0.55 | 0.78 |
| SO$_4^{2-}$ | 0.89 | 1.1 | 2.3 | 1.8 | 2.0 | 4.9 | 3.5 | 2.4 |
| Oxalate | 0.08 | 0.08 | 0.16 | 0.15 | 0.11 | 0.15 | 0.17 | 0.18 |