# Peer review of "Investigating the contribution of grown new particles to cloud condensation nuclei with largely varying preexisting particles – Part 1: Observational data analysis"

_EGUsphere, 2023_

## Author Response (AR1)

**Dear Editor,**

We revise our responses accordingly to the final revision. Please refer to the revised response. Thanks.

Your sincerely,

Xiaohong

10   Prof. Xiaohong Yao (Ph.D)

Ocean University of China

**Response to Reviewer 1**

*The manuscript discusses the impact of new particle formation on CCN based on a*
15   *measurement campaign conducted on a mountain site situated in the North China*
*Plain. The subject matter is significant, and the dataset is valuable, making me eager*
*to see a comprehensive study that is worthy of publication. However, I have concerns*
*about the manuscript's quality and the inadequate discussion of the findings. At its*
*present state, I cannot recommend it for publication in ACP. Major revisions are*
20   *necessary before it can be considered for publication. Please refer to my comments*
*and suggestions below.*

**Response:** The authors sincerely express gratitude for the constructive comments and have revised the manuscript accordingly. We acknowledge the importance of conducting a comprehensive comparison with previously reported mountain
25   observations and have incorporated this aspect to further enhance the depth of our analysis. Moreover, the weakness of these studies has also been clarified, particularly for their influences on our analysis.

*General comments:*

*1. In this study, the kappa parameter is used as a crucial link between CCN and*
30   *aerosol chemical composition in the analysis of the impact of NPF on CCN. However,*
*the computed kappa values in this study appear unusual. For instance, Figs. 3–5 often*

*displays a consistent low value below 0.1. Moreover, the equation (1) for calculating kappa cannot be applied if the value obtained is below 0.2 (Petters and Kreidenweis 2007). Additionally, for the situation examined in this study, the aerosol particles are likely to be externally mixed due to the presence of newly formed and pre-existing*
5 *particles. As a result, the relationship between CCN and kappa, assuming an internally mixed state, may exhibit significant deviations (Wex et al., 2010). It is crucial to discuss how this will affect the findings in this study.*

**Response:** The authors would like to express their appreciation for the comments received. Based on long-term Hygroscopic Tandem Differential Mobility Analyzer
10 measurements at a rural site (Wangdu, $38°40'$ N, $115°08'$ E, 51 m a.s.l.) in North China Plain (as conveyed in a private talk with Prof. Nan Ma), the annual averages of $\kappa$ values at the rural site indeed decreased from approximately 0.3 to around 0.15 during the last decade because of large decreases in $SO_2$ and $NO_x$ emissions in the region. Miao et al., (2015) also used a Hygroscopic Tandem Differential Mobility
15 Analyzer to measure $\kappa$ values of aerosols in the clean rural atmosphere at the top of Mt. Huang in China. They also found two distinct modes of $\kappa$ values, i.e., at approximately 0.3 and less than 0.1, respectively. A lot of $\kappa$ values in this study below 0.1 might reflect the real situation after substantial decrease in $SO_2$ and $NO_x$ emissions in NCP.

20 We agree that the calculated $\kappa$ values using Eq. (1) might suffer from the large errors. However, to compare with those widely reported in the literature, e.g., a long-term observation in 2008–2017 at the Cape Verde Atmospheric Observatory with the calculated $\kappa$ values from < 0.1 to 0.8 reported by Gong et al. (2022) from Leibniz-Institute for Tropospheric Research (Atmos. Chem. Phys., 22, 5175–5194, 2022, see
25 their Figs. 3–4), we calculated the $\kappa$ values larger than 0.1 and assigned a constant value of 0.05 to those below 0.1. Moreover, we adopted the approach used by various groups in research community to estimate $\kappa$ values. The approach assumes that the atmospheric aerosols are internally mixed. Nevertheless, this approach becomes invalid when the grown newly particles are large enough to be activated as CCN in
30 competing with pre-existing particles. The weakness and related references have been added in the revised version of the manuscript.

*2. The quality control of the CCN and PNSD dataset in this study appears to be unclear, which raises doubts about the reliability of the findings. Given that bulk CCN measurements were conducted on a mountain site, calibration of the CCN counter at*
35 *this altitude and correction for water depletion are necessary (Lance et al., 2006; Rose et al., 2008; Lathem and Nenes, 2011). Furthermore, because the critical diameter and kappa values are computed based on the comparison between PNSD and CCN number concentration, it is essential to verify the consistency between the CCN counter and CPC. For example, it is crucial to check how they perform when*
40 *measuring particles that are large enough for CCN activation under specific SS*

*conditions. Unfortunately, I could not find any information about these crucial aspects in this study.*

**Response:** To address the comments, a correlation analysis between $N_{ccn}$ at 0.2 % SS and $N_{cn>100}$ was conducted and have been added in the revised Supporting Information. The results of this analysis demonstrated these two variables were reasonably consistent. In fact, an even better correlation between $N_{ccn}$ at 0.2 % SS and $N_{cn>100}$ was observed during some short periods as presented in the text, e.g., "This was demonstrated by a significant correlation between $N_{ccn}$ and $N_{cn>100}$ from 10:00 to 14:00 on that day, with an equation of $N_{ccn} = N_{cn>100} \times 1.42 - 5.6 \times 10^2$, $R^2 = 0.83$, and $p < 0.05$ at 0.2 % SS." in Page 12, L15–17.

The FMPS had undergone maintenance at the TSI factory in the U.S. before the campaign, and the ratios of the measured total particle number concentrations against those measured by the CPC (CPC, TSI, 3775) remained stable until the end of 2021. This has been added in the revision. Moreover, the particle number size distributions (*PNSD*) measured by the FMPS have been corrected using the data simultaneously measured by the CPC according to the well-established approach in the literature. The accuracy of the total particle number concentrations measured by the CPC is critical for the correction. The inter-comparison of two identical CPCs (CPC, TSI, 3775) were also conducted after the campaigns because we had no two identical CPCs during the campaign. The after-campaign comparative results were highly consistent as presented in Supporting Information.

The measured *PNSD* is also reasonably consistent with those measured by a SMPS, although the SMPS had the constant underestimation because of particle loss. The comparison has been added in revised Supporting Information. Moreover, a comparison between the SMPS data and the wide-range particle sizer (WPS) data was conducted after the campaign, in which the two sizers showed a consistent *PNSD*, except the constant underestimation in the SMPS. The comparison results have also been added in Supporting Information.

We bought a commercial service provided by a DMT vendor in Beijing for the calibration of the CCN counter immediately before the campaign. We didn't make the on-site calibration for the CCN counter again. Based on the references, the absence of on-site calibration may lead to as large as ± 5 % uncertainty on the supersaturation (SS) and consequently cause up to 18 % analytic errors on the measured $N_{ccn}$. The information has been added in the revision. The analytic errors, however, less likely affect our comparison of $N_{ccn}$ at different times within a single day, when it can be reasonably assumed that analytic errors in percentage are invariant. This has clarified in the revision.

*3. In this study, the impact of both newly grown and pre-existing particles on CCN number concentration during NPF events was analyzed. It was found that in two NPF*

*cases, pre-existing particles, rather than newly grown particles, were responsible for the enhancement of CCN number concentration. This phenomenon is significant and may have occurred in other measurement activities, but was identified as NPF events that did not affect CCN number concentration. Since this phenomenon was observed*
5 *in two out of ten NPF cases in this study, it is worth investigating whether it has been observed in previous measurement studies. I recommend that the authors conduct a more thorough survey, including a comprehensive comparison with previous results, especially those obtained from mountain sites. This can significantly increase the novelty and significance of this study since if the enhancement of CCN number*
10 *concentration is incorrectly attributed to newly grown particles, the estimated contribution of NPF to CCN number concentration may be significantly skewed.*

**Response:** Agree. The comparison has been added in the revision, although the observations of CCN during the NPF events at the mountain sites are very limited. Moreover, Ren et al. (2021) recently conducted a comprehensive review of NPF
15 effects on CCN from 35 sites worldwide. The updated information has also been included in the revision. Please see revised P14, L14–40 and P15, L1–23 for further details.

*Specific comments:*

*1. L30–40 in P2: It would be beneficial to first introduce the global average kappa*
20 *before evaluating the kappa value of different chemical compositions as higher or lower.*

**Response:** Agree. In the revision, it reads as "Considering that the global average of the hygroscopicity parameter (κ) values of atmospheric aerosols is around 0.27 (Petters and Kreidenweis, 2007; Kerminen et al., 2012), the condensation of low-
25 volatile organic vapors on grown new particles and pre-existing particles might lead to a decrease in aerosol hygroscopicity due to their low κ values, which approach 0.1 (Petters and Kreidenweis, 2007; Dusek et al., 2010; Wu et al., 2013; Zhu et al., 2019b; Fang et al., 2021; Chang et al., 2022). However, some semi-volatile organic species, such as oxalic acid,…" in Page 2, L32–37.

30 *2. L31–33 in P7: Which rural site is being referred to? Is it the SMEAR II station? How does the comparison of PNSD and kappa between the SMEAR II station and the site in this study differ? Will these differences affect the comparison of the contribution of different sources to CCN in this study?*

**Response:** In the revision, it has been clarified to compare with the observations
35 made on Mt. Huang in China during the summer of 2014 and two other remote mountain sites made in the summer 2019 or 2020. Using the three mountain summer measurements as the upper of natural contributions, we can reasonably estimate the lower limit of the contribution from anthropogenic contributions.

In addition, Mt. Huang is surrounded by developing areas and less affected by anthropogenic air pollutants. In addition, their Hygroscopic Tandem Differential Mobility Analyzer observations on Mt. Huang showed the $\kappa$ values had two distinct modes distributions at ~ 0.3 and < 0.1, which are reasonably consistent with those estimated in this study.

*3. L20–21 in P10: Since the PNSD width of newly formed particles at a specific time can range up to tens of nanometers and the number concentration of newly formed particles that are much larger than the median mode diameter can still be higher than the number concentration of pre-existing particles, it is not appropriate to use the median mode diameter alone to determine whether the grown new particles were too small.*

**Response:** Agree. The width of *PNSD* should be considered as well as the median mode diameter. In the revision, it reads as "However, the grown new particles were still too small to be activated as CCN at 13:00–15:00 with the median mode diameter plus the corresponding half width of 24–30 nm in 99.7 % confidence to be considered. From 18:00 to 24:00, the maximum median mode diameter of the grown new particles stopped at 48 nm and the corresponding half width of 47 nm in 99.7 % confidence." in Page 11, L22–25.

*4. L2–3 in P13: As you have identified a more reasonable start time of estimating the net contribution of the grown new particles, please present your findings based on the 2.3analysis of the net contribution of the grown new particles.*

**Response:** The part has been revised as "However, using $N_{ccn,diff}$ at 11:00 appeared to be more reasonable than using $N_{ccn,diff}$ at 14:00 to estimate the net contribution of the grown new particles to $N_{ccn}$ at 1.0 % SS. The results of the test are presented below. We assumed the $N_{ccn,diff}$ value at 11:00 (864 cm$^{-3}$) to represent the $N_{ccn,diff}$ of pre-existing particles after 12:00, and assumed that the $N_{ccn,diff}$ was invariant after 12:00. It can obtain that the net contribution of the grown new particles was 769 ± 514 cm$^{-3}$ from 12:00 on 1 July to 05:00 on 2 July, accounting for only 23 ± 12 % of $N_{ccn}$ at 1.0 % SS. The maximum net contribution was 1.9 × 10$^3$ cm$^{-3}$ at 18:00 on 1 July, which accounted for 42 % of $N_{ccn}$ at 1.0 % SS. We also observed a minimum contribution of 4 % at 03:00 on 2 July, which was consistent with the disappearance of new particle signals. Alternatively, we tried that the $N_{ccn,diff}$ at 14:00 (1533 cm$^{-3}$) represented the substrate constant $N_{ccn,diff}$ of the pre-existing particles after 15:00. We observed negative net contributions of the grown new particles to $N_{ccn}$ at 1.0 % SS after 22:00 on July 1, suggesting that the $N_{ccn,diff}$ of pre-existing particles was overestimated." in Page 14, L2–14.

*5. L21–22 in P13: The references you have cited refer to measurements taken in polluted urban areas, which is not at all applicable to this study.*

**Response:** Thanks. The references have been updated in the revision.

*6. L31–37 in P13: The authors speculate that the formation of NH₄NO₃ and the condensation of hygroscopic organics are major drivers in increasing the CCN number concentration during NPF events, based on the measurement of NO₃ and SOC. However, it is important to consider the potential contribution of organonitrates, which are an important secondary aerosol composition consisting of both NO₃ and organics (Rollins et al., 2012; Ehn et al., 2014). This should be addressed in the discussion, as it may have an impact on the conclusions drawn in this study.*

**Response:** We agree that organonitrates are an important secondary aerosol composition, especially during nighttime. However, the hygroscopic properties of organonitrates are poorly characterized in the existing literature. We doubt whether they are comparable with $NH_4NO_3$ on affecting $\kappa$ values of atmospheric particles. Thus, we added the related discussion with secondary organic tracers in the revision. It reads as "In the literature, organonitrates were reported as an important secondary aerosol composition at nighttime. However, the species were not measured in this study. Thus, the influence of organonitrates on $\kappa$ values of the observed atmospheric particles is unknown." in Page 16, L33–35.

*7. L11–15 in P14: The authors compared the calculated CDNC to the measured CCN number concentration and concluded that only a small portion of CCN can form cloud droplets. While this conclusion may be correct, the authors' analysis is flawed. This is because CDNC is determined not only by the number of CCN but also by the actual SS present in the atmosphere (Pruppacher and Klett, 2012). Under varying meteorological conditions, the SS can differ significantly and can significantly impact CDNC even if the number of CCN remains constant. Therefore, this aspect needs to be taken into account in the corresponding discussions.*

**Response:** Agree. The part has been revised as "The *CDNC* is a strong function of the actual SS present in the atmosphere in addition to $N_{ccn}$ (Pruppacher and Klett, 2012). This large difference between the observed $N_{ccn}$ and satellite-derived *CDNC* implies that the actual SS in the atmosphere might be substantially smaller than 0.2 %. Shen et al. (2018) recently reported the actual SS varying from 0.01 % to 0.05 % during fog events observed in the NCP. Thus, it is not surprising to find that only a small fraction of CCN could competitively capture water vapor to form cloud droplets during the study period (Shen et al., 2018; Jiang et al., 2021; Gong et al., 2023). Moreover, the *CDNC* during the period from 29 June to 14 July was $120 \pm 86$ cm$^{-3}$ and close to the seasonal average in June and July. The satellite-derived *CDNC* on 2–3 July were even lower than the average, suggesting that the NPF event was unlikely to have any influence on *CNDC* at such low actual SS." in Page 16, L39–Page 17, L9.

*8. L7–8 in P16: This speculation is difficult to understand. In addition, please provide more information on the implications or prospects on the findings of this study. In*

*particular, given that you have a companion paper, presenting it and discussing its relevance will highlight the importance of this research.*

**Response:** It revised as "This suggests that the actual SS required to form cloud droplets in the NCP atmospheres may be substantially smaller than 0.2 %. Again, the grown new particles didn't yield a detectable contribution to $N_{ccn}$ at 0.2 % SS during all eight NPF events. Thus, it is reasonably argued that the grown new particles might not act as $N_{ccn}$ and form cloud droplets in NCP atmospheres with actual SS largely smaller than 0.2 %." in Page 19, L10–L14.

**Technical corrections:**

*1. L20 in P4: It should be "splitter" rather than "spitter".*

**Response:** Corrected.

*2. L36 in P4: What are the start and end times of the TSP sampling?*

**Response:** The information has been added in the revision. It started from 06:00 on that day and ended at 06:00 on the next day.

*3. L16–19 in P8: This sentence is unclear. Please revise for clarity.*

**Response:** In the revision, it reads as "However, the occurrence of pre-existing particle growth seemed infrequent and was observed only on 4 July (1 of 16 days). On that day, larger and smaller pre-existing particle growth was observed from 88 nm to 116 nm and from 24 nm to 32 nm, respectively (See Fig. S7)." in Page 9, L15–L18.

*4. L39–40 in P8: This sentence is unclear. Please revise for clarity.*

**Response:** For clarification, the last two sentences are combined in the revision. It reads as "It is important to note that the observational data alone cannot provide evidence of any additional evolution of grown new particles and their additional contribution to $N_{ccn}$ after the new particle signal disappears, particularly considering the infrequent occurrence of the pre-existing particle growth." in Page 9, L36–L39.

*5. L25 in P10: It's odd to say "organic vapor was growing on the pre-existing particles". Use "condensing" instead.*

**Response:** Corrected.

*6. L3 in P15: There is no definition of PC1 or PC2. Please provide a definition.*

**Response:** The PC has been defined as: "The selected peak spectral Principal Component Analysis (PCA) of SIMS data was also analyzed to identify variations among the samples. Scores plots were generated to show the similarities and dissimilarities among the samples. The most important principal component (PC1) and the second important principal component (PC2) explained 84.5 % and 9.1 % of the data, respectively." in Page 18, L2–L5.

*7. Fig. 2: The range of the Y-axis in panel (c) is too large, and it appears that there is no variation of CDNC with time. Please adjust the range of the Y-axis.*

**Response:** Corrected.

*8. Figs. 3–5: Please include a time series of the critical diameter (Dc) in the panel of PNSD.*

**Response:** Added.

*9. Fig. 7: The markers in panel (a) are too small to identify. Please increase their size.*

**Response:** Corrected.

***References:***

*[1] Ehn, M., Thornton, J. A., Kleist, E., et al.: A large source of low-volatility secondary organic aerosol, Nature, 506, 476–479, https://doi.org/10.1038/nature13032, 2014.*

*[2] Lance, S., Nenes, A., Medina, J., and Smith, J. N.: Mapping the operation of the DMT continuous flow CCN counter, Aerosol Sci. and Technol., 40, 242–254, https://doi.org/10.1080/02786820500543290, 2006.*

*[3] Lathem, T. L. and Nenes, A.: Water vapor depletion in the DMT continuous-flow CCN chamber: Effects on supersaturation and droplet growth, Aerosol Sci. and Technol., 45, 604–615, https://doi.org/10.1080/02786826.2010.551146, 2011.*

*[4] Petters, M. D. and Kreidenweis, S. M.: A single parameter representation of hygroscopic growth and cloud condensation nucleus activity, Atmos. Chem. Phys., 7, 1961–1971, https://doi.org/10.5194/acp-7-1961-2007, 2007.*

*[5] Pruppacher, H. R. and Klett, J. D.: Microphysics of clouds and precipitation: Reprinted 1980, Springer Science & Business Media, Berlin, Germany, 2012.*

*[6] Rollins, A. W., Browne, E. C., Min, K. E., et al.: Evidence for $NO_x$ control over nighttime SOA formation, Science, 337, 1210–1212, https://doi.org/10.1126/science.1221520, 2012.*

*[7] Rose, D., Gunthe, S. S., Mikhailov, E., et al.: Calibration and measurement uncertainties of a continuous-flow cloud condensation nuclei counter (DMT-CCNC): CCN activation of ammonium sulfate and sodium chloride aerosol particles in theory and experiment, Atmos. Chem. Phys., 8, 1153–1179, https://doi.org/10.5194/acp-8-1153-2008, 2008.*

*[8] Wex, H., McFiggans, G., Henning, S., and Stratmann, F.: Influence of the external mixing state of atmospheric aerosol on derived CCN number concentrations, Geophys. Res. Lett., 37, 10805–10808, https://doi.org/10.1029/2010gl043337, 2010.*

**Response:** The refences have been included in the revision to improve the quality of discussion.

**Response to Reviewer 2**

*General comments*

*This paper is a case study discussing whether new particles generated by new particle formation (NPF) events grow and contribute to CCN based on two weeks of atmospheric observations. It is an important topic involving aerosol-cloud interactions, and the accumulation of such case studies is meaningful given the large spatiotemporal variability of aerosol properties. However, because of the following problems, a major revision of manuscript is needed before it can be accepted for publication.*

*The authors explain the case in several categories as to whether NPF events contribute to CCN number concentrations, but it feels illogical that the method of categorization is ambiguous and that even on the same day it is discussed in separate sections by supersaturation. In addition, cloud particle number concentrations were not measured in the observation site, so caution should be exercised when comparing calculated values with observed CCN number concentrations. The authors raise three research questions for introduction, so in conclusion part, it is necessary to answer these clearly. In particular, there is no description of the answer to the third research question: What implications do our findings have on knowledge gaps for CCN sources in NCP?*

**Response:** The authors carefully considered the comments and revised accordingly to enhance the overall quality of the presentation.

*Other detailed points are listed below:*

*Specific comments*

*1. P2 L16 The lower limit of the specific particle size in this study should be mentioned in the method section.*

**Response:** The part has been revised as "The grown new particles have been reported to contribute to the budget of CCN according to field measurements and modeling studies (Kuang et al., 2009; Kerminen et al., 2012; Ehn et al., 2014; Kalivitis et al., 2015; Leng et al., 2014; Ma et al., 2016; Tröstl et al., 2016; Gordon et al., 2017; Li et al., 2017b; Williamson et al., 2019; Fang et al., 2021; Sebastian et al., 2021)." in Page 2, L19–L22.

*2. P3 L28 Can the authors identify or estimate the period of cloud coverage at the observation site? Are data on relative humidity in the atmosphere, for example, available? Because cloud formation contributes to the deposition of pre-existing*

*particles, it is considered an important data to deepen the discussion of NPF case studies.*

**Response:** It is difficult to accurate estimate the period of cloud coverage at the mountain area since the cloud coverage highly varied from time to time and had a large spatial inhomogeneity. In the revision, satellite cloud data and the ground-level relative humidity were presented in the Supporting Information to facilitate the analysis. It reads "The observations were conducted at a rural mountain site (1100 m above sea level) in North China Plain (NCP) from June to July in 2019, during the rainy season when morning mist frequently occurred among mountain peaks with ambient relative humidity approaching to 90 %–100 % and decreasing ambient temperature (Fig. S1a–b). Additionally, cloud optical thickness (*COT*) and cloud effective radius (*CER*) were recorded in the midafternoon using Aqua satellite data, showing variations from 1.46 to 36.7 and 8.64 μm to 24.4 μm, respectively (Fig. S1c–d). The observed CCN were directly related to the formation of cloud droplets at the elevated mountain site." in Page 3, L26–L32.

*3. P4 L12 It would be helpful if the authors include the piping diagrams for the four instruments in the supplement material.*

**Response:** The corresponding piping diagrams has been added as Fig. S2 in the supporting information.

*4. P4 L22 How often switch the SS setting for CCNC? i.e., How many minutes each supersaturation setting lasted?*

**Response:** As clarified in the revision, each supersaturation (SS) setting for the CCNC lasted for a duration of 5 minutes, with the first and last 30 seconds of data being removed. However, an additional 5 minutes were used for switching SS from 1.0 % to 0.2 % to establish supersaturation equilibrium.

*5. P6 L6 Yao et al. (2005) is missing in the reference list. It should be Yao et al. (2007) or Yao et al. (2010)?*

**Response:** Sorry for this. It has been corrected in the revision.

*6. P6 L26 It is questionable whether these satellite products reflect clouds that form at observation sites. In the introduction, the authors write that observation sites are often covered in morning mist, but can satellite products capture this phenomenon?*

**Response:** Satellite data reflect observations between 12:30–14:30. This has been clarified in the revision. However, the on-site ambient relative humidity can indirectly support the observed phenomenon (please see our revision to Question 2).

*7. P6 L29 How many days back in the backward trajectory analysis?*

**Response:** 24-hour air mass back trajectories were used for this study. This has been added in the revision. 24-hour air mass back trajectories rather than even longer time trajectories were used because of lack of strong anthropogenic $SO_2$ emissions in the remote northwest and north areas. Thus, the $SO_2$ in low concentration levels was expected to be mostly removed or oxidized after the 24-hour long-range transport in the atmosphere.

*8. P7 L18 As commented on the methodology, it is questionable whether the CDNC calculated based on the satellite products reflects the CDNC of the lower clouds at the observation site. As the CDNC strongly depends on the water vapor supersaturation in the cloud and, for that matter, the ascent velocity of the air mass, so it is also advisable to avoid comparing CCN concentration with CDNC without these discussions.*

**Response:** We agree the actual SS mainly determined by ascent velocity of the air mass is critical in continental atmospheres affected by anthrophonic air pollution to some extent. In the revision, we added related discussion accordingly. It reads as "These values were higher compared to the CDNC levels observed in June–July of 2019, which varied around $141 \pm 122$ cm$^{-3}$ (fig superimposed in Fig. 2d). On the other 14 days, the CDNC were determined to be $106 \pm 55$ cm$^{-3}$, approximately 1 order of magnitude smaller than the observed corresponding to $N_{ccn}$ at 0.2 % SS. The difference between the observed $N_{ccn}$ and the $CDNC$ implied that the SS required for cloud droplet formation in the NCP atmospheres might be substantially smaller than 0.2 %, even though $N_{ccn}$ may decrease to some extent at higher altitudes from 1000 m (Li et al., 2019; Yang et al., 2020; Che et al., 2021). Thus, we will further explore this difference in Sect. 3.5 Note that the estimation $CDNC$ might suffer from the uncertainty to some extent because of complex microphysics of cloud (Pruppacher and Klett, 2012). However, this uncertainty is unlikely to have significantly affected the comparison results of $CNDC$ with $N_{ccn}$ since the latter difference generally exceeded 1 order of magnitude." in Page 7, L32–Page 8, L4.

Moreover, the grown new particles didn't yield a detectable contribution to $N_{ccn}$ at 0.2 % SS during all eight NPF events in this study. Thus, it is reasonably argued that the grown new particles might not act as $N_{ccn}$ in NCP atmospheres with actual SS largely smaller than 0.2 %. This has been added in the revised Sect. 3.4.

*9. P7 L24 It's not mandatory but there is a paper that summarizes CCN number concentrations (at 0.2 % SS) at sites around the world (including mountain sites). Schmale, J.et al.: Long-term cloud condensation nuclei number concentration, particle number size distribution and chemical composition measurements at regionally representative observatories, Atmos. Chem. Phys., 18, 2853–2881, https://doi.org/10.5194/acp-18-2853-2018, 2018.*

**Response:** Agree. We have added a comparison of the observations of CCN during the NPF events at the mountain sites in the revision. Please see revised part in P8, L5–29.

*10. P7 L31 This reviewer feels there is a weak basis for estimating the contribution rates of natural and anthropogenic sources. Since the observation site is in the vicinity of Beijing, it may depend strongly on the wind directions.*

**Response:** This is a rough estimation. When the wind directions from the south and southwest, the contribution of $N_{ccn}$ from primary and secondary anthropogenic aerosols should be even larger. In the revision, we added "In Beijing, NPF events have also been observed in polluted atmospheres with air masses originating from the south and southwest (Wu et al., 2007). In those cases, anthropogenic aerosols expectedly yield an even larger contribution to $N_{ccn}$." in Page 8, L27–29.

*11. P8 L30 This paragraph barely discusses particle size. Whether or not a particle acts as a CCN depends strongly on the particle size, so it is natural that a particle in the newly formed nucleation mode does not contribute to $N_{CN>100}$ or $N_{CCN}$.*

**Response:** Agree. This is why we presented the case-by-case examination of the growth of new particles and the potential contribution to $N_{ccn}$. To better service the reader, we revised the paragraph immediately after. It reads as "It is important to note that the observational data alone cannot provide evidence of any additional evolution of grown new particles and their additional contribution to $N_{ccn}$ after the new particle signal disappears, particularly considering the infrequent occurrence of the pre-existing particle growth." in P9, L36–39, and "Therefore, this paper presents a case-by-case examination from Sect. 3.2 to Sect. 3.5 where the contributions of NPF events to the $N_{ccn}$ are elaborated at various SS with considering the grown new particles in different sizes." in Page 10, L5–7.

*12. P9 L5 Wang Y., et al., 2020 seems to be missing in the reference list.*

**Response:** It is "Wan et al., 2020". Corrected.

*13. P9 L11 This reviewer doesn't quite understand the intent of discussing the events of July 1 in subsections by supersaturation. Shouldn't 3.4 and 3.5 be in the same section?*

**Response:** The two subsections have been combined.

*14. P9 L24 Because of the wide range of kappa value of organics, this is likely to depend largely on the type of organic matter.*

**Response:** In the revision, it reads as "Assuming that the activated aerosols at 0.2 % SS were internally mixed and mainly composed of inorganic ammonium salts and organics (Petters and Kreidensohler, 2007; Rose et al., 2010, 2011), both of them likely yielded an appreciable contribution to the total mass concentration of the associated aerosols. However, the exact percentages relied on the $\kappa$ values of various organics." in Page 10, L24–27.

*15. P10 L36 The 3 July case also confirmed an increase in $N_{CCN}$, but concluded that this was due to pre-existing particles. this reviewer is not sure what the difference is between this case (3 July) and the section 3.3 cases. Couldn't the case of 3 July be included in 3.3?*

**Response:** We have taken the comments into careful consideration. On the mentioned day, the $N_{ccn}$ generally decreased during most of the NPF period. However, $N_{ccn}$ indeed largely increased from a lower level to a higher level in 2 hours of the NPF event. Since the period with decreasing $N_{ccn}$ was much longer than the 2 hours with increasing $N_{ccn}$, we have chosen to present the 3 July case in Sect. 3.2 rather than Sect. 3.3. This complex situation is not very surprised by considering spatial inhomogeneity of pre-existing $N_{ccn}$. We thank the reviewer's comment and hope that the reviewer can agree with us on this issue.

*16. P14 L8 Same comment as P7–L18.*

**Response:** The paragraph has been revised as "The *CDNC* is a strong function of the actual SS present in the atmosphere in addition to $N_{ccn}$ and the actual SS is determinated by the ascent velocity of the air mass, the amount of moisture, etc (Pruppacher and Klett, 2012). This large difference between the observed $N_{ccn}$ and satellite-derived *CDNC* implies that the actual SS in the atmosphere might be substantially smaller than 0.2 %. Shen et al. (2018) recently reported the actual SS varying from 0.01 % to 0.05 % during fog events observed in the NCP. Thus, it is not surprising to find that only a small fraction of CCN could competitively capture water vapor to form cloud droplets during the study period (Shen et al., 2018; Jiang et al., 2021; Gong et al., 2023). Moreover, the *CDNC* during the period from 29 June to 14 July was $120 \pm 86$ cm$^{-3}$ and close to the seasonal average in June and July. The satellite-derived *CDNC* on 2–3 July were even lower than the average, suggesting that the NPF event was unlikely to have any influence on *CNDC* at such low actual SS.

When the *COT* and *CER* were compared during NPF days and non-NPF day, the former *COT* values around $14.2 \pm 5.9$ had no significant difference from the latter values around $17.2 \pm 11.8$ with $p = 0.53$. The same was true for the *CER* values, i.e., $13.2 \pm 3.1$ μm during NPF days versus $16.8 \pm 4.0$ μm during non-NPF days with $p = 0.067$. Lack of significant differences on the two cloud parameters between NPF days and non-NPF days might be related to the small dataset." in Page 16, L39–Page 17, L14.

*17. Section 3.7 This part is unique and interesting but somewhat speculative. Can't SIMS get information about inorganic materials on the particle surface? If possible, discussing it in conjunction with organic matter would make the discussion more robust.*

**Response:** SIMS can detect inorganic ions, but the interference is large because of the weak ratios of signal to noise. This has been clarified in the revision. We have to limit our analysis on organic components with high ratios of signal to noise.

*18. P14 L34 This reviewer does not understand why the authors concluded that condensation of inorganic vapors is dominant in larger size particles (> 60 nm).*

**Response:** SIMS used in this study suffers from the weakness on analyzing in-depth chemical components of atmospheric particles. Due to smaller Kelvin effects, semi-volatile organic and inorganic vapors may have overwhelmingly condensed on the sized particle surfaces with diameters larger than 60 nm. Considering the amounts of semi-volatile organic and inorganic vapors in the atmosphere is substantially larger than that of low-volatile vapors, the semi-volatile compound signals on > 60 nm particles may cover up the signals of high-molecular-weight organics in low volatility. The sentence has been revised as "when semi-volatile organic and inorganic vapors may have overwhelmingly condensed on the sized particle surfaces and covered up the high-molecular-weight organic fragment signals." in Page 17, L34–L36.

*19. Figure 1 Airmass on the day of the NPF event is being transported from the north or northwest direction. Was there a difference in the airmass pathway on days with and without NPF events?*

**Response:** Wu et al. (2007) reported a statistical analysis of NPF events using a year-long measurement in Beijing, i.e., NPF events were more likely to occur under low RH and sunny conditions while non-NPF events were usually associated with strong condensational sink or absence of sunny conditions. The air masses from the north and northwest usually carried dry and clean air, favoring the occurrence of NPF events. However, Wu et al also reported that some NPF events were associated with air masses from the south and southwest. The reference has been added and summarized to support the analysis in the revision. In this study, our observational period was too short to statistically analyze the difference in air mass pathways between days with and without NPF events.

*20. Figure 2a It is very hard to see and misleading if separate axes for $N_{CN}$ and $N_{CCN}$ were used. Can it be represented by a single axis, with the vertical axis being the logarithmic axis? Also, making a separate graph of the activation ratio ($N_{CCN}/N_{CN}$) would be helpful to see how the cloud activation potential is.*

**Response:** Revised and added accordingly.

*21. Figure 2c Please zoom in on the vertical scale to see the changes more easily. What is the meaning of the inset figure (CDNC in June and July)?*

**Response:** Revised. The observational period alone is too short to be representative. The two month-long data of *CDNC* can better reflect its seasonal variability. This has been clarified in the revision.

*22. Figure 3b and 3e Same comment as Figure 2a. Better to have $N_{CN>100}$ and $N_{CCN}$ on the same axis.*

**Response:** Revised accordingly.

*23. Figure 4b and 4e Same comment as above. Also, the captions of the figures should provide sufficient explanation to understand them. (without looking at the text or other figures)*

**Response:** Revised accordingly.

*Matters related notations*

*1. "P" for p-values should be italicized in lower case letter "p".*

*2. All variables ($N_{CN}$, $N_{CCN}$, $\kappa$....) in the manuscript and supplementary material should be italicized.*

**Response:** Corrected.

---

## Author Response (AR2)

**Dear Editor,**

We would like to express our gratitude for the comments provided by the reviewer. We have revised our manuscript to address these comments and improve the quality of our work. Your patience and assistance throughout the review process are greatly appreciated.

Your sincerely,

Xiaohong

Prof. Xiaohong Yao (Ph.D)

Ocean University of China

**Response to Major comments:**

*1. On-site calibration was not conducted during this campaign. However, the authors posit that there was merely a ±5 % uncertainty in the determination of supersaturation (SS), leading to potential analytic errors of up to 18 % in the measured $N_{ccn}$. However, the influence of ambient pressure variations on SS in CCN counter is similar to that of temperature difference between the top and bottom of the CCN counter column (Eq. (16) in Lance et al., 2006; Fig. 5d and 8c in Rose et al., 2008). The divergence in height between the observation site and the calibration location can significantly perturb the SS within the CCN counter. Specifically, considering that the site's elevation is roughly 1000m, resulting in a potential pressure decrease of around 10 % compared to that of Beijing city, employing calibration outcomes from Beijing city could lead to a 10 % systematic overestimation of SS in CCN measurements at this particular site, even if we overlook variations in calibration between locations at the same elevation. If, as stated by the authors, an 18 % error can arise due to a ±5 % uncertainty in SS, thus the overestimation of SS by 10 % might consequently result in errors of up to 36 % in measured $N_{ccn}$. However, these deviations in measured $N_{ccn}$ can exhibit a non-linear pattern depending on variations in particle number size distribution and hygroscopicity distribution. This non-linear effect is also significant for the calculation of critical diameter and, subsequently, kappa values. Given the potential complexity of these issues, a meticulous examination is imperative. Neglecting this could cast doubt upon the credibility of the findings presented in this study.*

**Response:** We appreciate the valuable suggestions. In absence of on-site calibration, we agree that the use of 10 % reduction in supersaturation (SS) is a reasonable approximation at the mountain site when comparing with values derived from laboratory calibration in Beijing. Consequently, we have revised the manuscript accordingly.

In Page 4, lines 28–37, of the revised version, it reads "However, the SS values were referred as lab-calibrated values in this study since no on-site calibration was conducted during the campaign. The divergence in height between the observational site and the calibration location (~1000 m) may introduce uncertainties on SS and the consequently measured $N_{ccn}$ (Lance et al., 2006; Rose et al., 2008; Lathem and Nenes, 2011). Based on these previous studies, the on-site five SS at the mountain site might be approximately 10 % smaller than the corresponding lab-calibrated values, i.e., 0.18 %, 0.36 %, 0.54 %, 0.72 % and 0.9 %. These smaller SS values were referred as the approximated on-site values in this study. Furthermore, the errors in the measured $N_{ccn}$ between each pair of lab-calibrated and approximated on-site SS were found to be smaller than 10 %, as presented in Supporting Information (Figs. S6–S10)."

Text S4 in Supporting Information was newly added to analyze $N_{ccn}$ and κ values at lab-calibrated SS and the approximated on-site SS. Moreover, additional results were added in the main text to support the related analysis, by considering the approximated on-site SS.

**Response to Minor comments:**

*1. The authors acknowledge the potential for significant errors when calculating kappa values using Eq. (1). Eq. (1) is Eq. (10) in Petters and Kreidenweis (2007), which is an approximate expression of Eq. (6) in Petters and Kreidenweis (2007) when kappa values were larger than 0.2. Thus I suggest including a comparison between the kappa values calculated from these two equations (i.e., both Eq. (6) and Eq. (10) in Petters and Kreidenweis, 2007) in the supplementary materials. Incorporating a note of caution alongside the comparison within this paper will be also helpful.*

**Response:** Eq. (6) and Eq. (10) presented in Petters and Kreidenweis (2007) provide two methods for calculating $\kappa$ values. The Eq. (6) was the derivation of Kohler equation, which can be applied across entire range of supersaturation levels and solute hygroscopicity, resulting in smaller errors. However, this equation requires the growth factor obtained from HTDMA measurements, which were not available in this study. In fact, the absence of simultaneous HTDMA measurements was also the exact reason for most studies to use the Eq. (10).

*2. Page 11, Line 29: The term "less CCN-activated organic vapor " lacks clarity. It may lead to a misunderstanding that there is a reduced presence of "organic vapor which is CCN-activated" condensing on particles.*

**Response:** Agree. It reads as "It is possible that the growth of pre-existing particles was driven by organic vapor with lower CCN activation since κ values at 0.4 % SS decreased from 0.11 to lower values." in Page 11, L38–40.

*3. Page 12, L24-25: This sentence "Thus, at least 1 % of the grown new particles were large enough to be activated as CCN." seems to be problematic in its expression. The conclusion that "only 1 % of the newly grown particles can serve as cloud condensation nuclei" itself lacks meaningful significance.*

**Response:** In the revision, it reads as "Thus, at least 1 % of the grown new particles (61–73 cm⁻³) were large enough to be activated as CCN. Supposed that the part of grown new particles were totally activated as cloud droplets, they should yield an appreciable contribution to *CNDC*." in Page 12, L35–37.

*4. Page 16, Line 39 - Page 17, Line 9: A notable disparity exists between the observed $N_{ccn}$ and satellite-derived CDNC. This discrepancy is attributed to the likelihood that the actual SS in the atmosphere could be considerably less than 0.2 %. However, the SS value mentioned by the authors was obtained in a fog at a rural site within the NCP. While it's not guaranteed that the SS in clouds near the mountain site in this study is necessarily close to or higher than 0.2 %, it's also improbable for it to be close to SS values in fogs at significantly lower altitudes. Stronger evidence is necessary.*

**Response:** In the revision (Page 17, lines 18–29), we added "This large difference

between the observed $N_{ccn}$ and satellite-derived $CDNC$ implies that the actual SS in the atmosphere might be substantially smaller than 0.2 %. In fact, Gao et al. (2021) recently conducted aircraft observations over Beijing and calculated the SS at cloud base to be approximately 0.048 %. Moreover, Shen et al. (2018) also reported the actual SS values ranging from 0.01 % to 0.05 % during fog events observed in the NCP. Iwamoto et al. (2021) reported the mean SS around 0.34 % during cloud-shrouded periods at Mt. Fuji in Japan. Notably, their observed SS decreased to 0.24 % when the air mass originated from continental sources. The higher SS observed at Mt. Fuji might be related to substantially lower $N_{ccn}$ (around 108 cm$^{-3}$ at 0.21 % SS) than those in Beijing. The reduced effect on SS levels with increasing $N_{ccn}$ was also obtained in Oklahoma (Jia et al., 2019), in which the estimated SS of stratocumulus and cumulus in relatively polluted atmospheres approximately equaled to 0.2 %. However, the $N_{ccn}$ in their relatively polluted atmospheres were smaller than half of the observed $N_{ccn}$ in this study."

---

## Author Response (AR3)

**Dear Editor,**

Your continuous patience and invaluable assistance throughout the review process are deeply appreciated. We would like to express our gratitude for the comments provided by the reviewer. We have revised our manuscript to address these comments and improve the overall quality of our work.

Your sincerely,

Xiaohong

Prof. Xiaohong Yao (Ph.D)

Ocean University of China

**Response to comments:**

*1. In the revised manuscript, the figures in the main text exclusively present NCCN and kappa values without accounting for the deviation of the SS calibration, which is referred to as "lab-calibrated" in the manuscript. Conversely, NCCN and kappa values that take into consideration the deviation of the SS calibration, referred to as "approximated on-site" in the manuscript, are only depicted in the supplementary material. I believe that this presentation may potentially lead to misconceptions, suggesting that the "lab-calibrated" values are more accurate than the "approximated on-site" values. This is particularly noteworthy when it comes to estimating kappa because in cases where NPF due to the SS in the CCN counter results in overestimation, kappa could be underestimated by approximately 20%. Therefore, I encourage the authors to include both "lab-calibrated" and "approximated on-site" values of NCCN and kappa in the figures within the main text. Doing so would serve as a reminder to readers to exercise caution when interpreting the NCCN and kappa values reported in this study.*

**Response:** We appreciate the valuable suggestions, which are helpful enhancing the overall quality of our work. In our revision, both "lab-calibrated" and "approximated on-site" values of $N_{ccn}$ and $\kappa$ values for the case studies have been added into the figures within the manuscript and Supporting Information.

*2. In response to the minor comments from my previous review, the authors asserted that the utilization of Equation (6) depends on a growth factor derived from HTDMA measurements, which were unavailable in this study. They also mentioned that the absence of concurrent HTDMA measurements was the primary reason why most studies rely on Equation (10). However, this argument is incorrect. In Petters and Kreidenweis (2007), it is explicitly stated that this equation can be applied under conditions of cloud droplet activation without any mention of the necessity for a growth factor or HTDMA data: "Equation (6) applies over the entire range of relative humidity and solution hygroscopicity. It can thus be used to predict particle water content in the subsaturated (S<1) regime, as well as to predict the conditions for cloud droplet activation. The critical supersaturation (sc, where sc=Sc–1 and is usually expressed as a percentage) for a selected dry diameter of a particle having hygroscopicity κ is computed from the maximum of the κ-Kohler curve (Eq. 6). Figure 1 shows the relationship between dry diameter and critical supersaturation for a range of constant κ values, computed for σ s/a =0.072 J m−2 and T =298.15 K." Moreover, this equation has been employed in other research as well. For instance, in Kerminen et al. (2012), it was referred to as Equation (1) and utilized to calculate the relationship between particle dry size and critical supersaturation, as demonstrated in Figure 1 of that work, without any indication of the need for a growth factor or HTDMA data. I strongly urge the authors to rectify this matter.*

**Response:** We are sorry that our previous response on this issue was not clear enough and caused your concerns. We try to provide a more comprehensive explanation here:

Eq. (6) in Petters and Kreidenweis (2007) is derived from the Köhler theory, in which the hygroscopicity parameter $\kappa$ replaces the solution activity. However, Petters and Kreidenweis (2007) didn't use this method for $\kappa$ calculation. Instead, they employed a simple rule, Eq. (7), to get a range of constant $\kappa$ values, where $\kappa$ is defined as $\kappa = \sum \varepsilon_i \kappa_i$. Here, $\varepsilon_i$ represents the volume fraction of a certain chemical compound in aerosol and the $\kappa_i$ represents the hygroscopicity parameter of that compound. When utilizing Eq. (6) to calculate $\kappa$ values, the inclusion of a wet diameter ($D_{wet}$) or growth factor is necessity. These parameters need to be obtained from HTDMA measurements, a practice employed in numerous studies (e.g., Kawana et al., 2017; Cerully et al., 2011). Unfortunately, since HTDMA data was not available in our study, we selected for Eq. (10) from Petters and Kreidenweis (2007) for $\kappa$ value calculations.

Moreover, Fig. 1 in Professor Kerminen's research (Kerminen et al., 2012) shows the relationship between the calculated critical supersaturation ($S_c$) and dry diameter ($D_{dry}$), in which $\kappa$ values serve as constant input values.

In fact, there were three primary approaches to determine $\kappa$ values in the literature, i.e., 1) Using Eq. (7) as showed in Petters and Kreidenweis (2007); 2) The use of HTDMA measurements; and 3) Employing either directly measured critical diameters at a super saturation (Rose et al., 2010) or estimated ones (in our study).

We hope that our explanation can address your concerns.

*References:*

*[1] Kerminen, V. M., Paramonov, M., Anttila, T., Riipinen, I., Fountoukis, C., Korhonen, H., Asmi, E., Laakso, L., Lihavainen, H., Swietlicki, E., Svenningsson, B., Asmi, A., Pandis, S. N., Kulmala, M., and Petäjä, T.: Cloud condensation nuclei production associated with atmospheric nucleation: A synthesis based on existing literature and new results, Atmos. Chem. Phys., 12, 12037–12059, https://doi.org/10.5194/acp-12-12037-2012, 2012.*

*[2] Petters, M. D. and Kreidenweis, S. M.: A single parameter representation of hygroscopic growth and cloud condensation nucleus activity, Atmos. Chem. Phys., 7, 1961–1971, https://doi.org/10.5194/acp-7-1961-2007, 2007.*

*[3] Kawana, K., Nakayama, T., Kuba, N., and Mochida, M.: Hygroscopicity and cloud condensation nucleus activity of forest aerosol particles during summer in Wakayama, Japan, J. Geophys. Res.-Atmos., 122, 3042–3064, https://doi.org/10.1002/2016jd025660, 2017.*

*[4] Cerully, K. M., Raatikainen, T., Lance, S., Tkacik, D., Tiitta, P., Petäjä, T., Ehn, M., Kulmala, M., Worsnop, D. R., Laaksonen, A., Smith, J. N., and Nenes, A.: Aerosol hygroscopicity and CCN activation kinetics in a boreal forest environment during the 2007 EUCAARI campaign, Atmos. Chem. Phys., 11, 12369–12386, https://doi.org/10.5194/acp-11-12369-2011, 2011.*

[5] Rose, D., Nowak, A., Achtert, P., Wiedensohler, A., Hu, M., Shao, M., Zhang, Y., Andreae, M. O., and Pöschl, U.: Cloud condensation nuclei in polluted air and biomass burning smoke near the mega-city Guangzhou, China – Part 1: Size-resolved measurements and implications for the modeling of aerosol particle hygroscopicity and CCN activity, Atmos. Chem. Phys., 10, 3365–3383, https://doi.org/10.5194/acp-10-3365-2010, 2010.